# Concomitant control of mechanical properties and degradation in resorbable elastomer-like materials using stereochemistry and stoichiometry for soft tissue engineering

Mary Beth Wandel[1,7], Craig A. Bell [2,3,4,7], Jiayi Yu[1,7], Maria C. Arno [5], Nathan Z. Dreger [1], Yen-Hao Hsu[1], Anaïs Pitto-Barry [2], Joshua C. Worch [5], Andrew P. Dove [5✉] & Matthew L. Becker [6✉]

Complex biological tissues are highly viscoelastic and dynamic. Efforts to repair or replace cartilage, tendon, muscle, and vasculature using materials that facilitate repair and regeneration have been ongoing for decades. However, materials that possess the mechanical, chemical, and resorption characteristics necessary to recapitulate these tissues have been difficult to mimic using synthetic resorbable biomaterials. Herein, we report a series of resorbable elastomer-like materials that are compositionally identical and possess varying ratios of *cis:trans* double bonds in the backbone. These features afford concomitant control over the mechanical and surface eroding degradation properties of these materials. We show the materials can be functionalized post-polymerization with bioactive species and enhance cell adhesion. Furthermore, an in vivo rat model demonstrates that degradation and resorption are dependent on succinate stoichiometry in the elastomers and the results show limited inflammation highlighting their potential for use in soft tissue regeneration and drug delivery.

[1] Department of Polymer Science, The University of Akron, Akron, OH 44325, USA. [2] Department of Chemistry, The University of Warwick, Coventry CV4 7AL, UK. [3] Centre for Advanced Imaging, The University of Queensland, St Lucia, QLD 4072, Australia. [4] Australian Institute for Bioengineering and Nanotechnology, The University of Queensland, St Lucia, QLD 4072, Australia. [5] School of Chemistry, The University of Birmingham, Edgbaston, Birmingham B15 2TT, UK. [6] Department of Chemistry, Mechanical Engineering and Materials Science, Orthopaedic Surgery, Duke University, Durham, NC 20899, USA. [7] These authors contributed equally: Mary Beth Wandel, Craig A. Bell, Jiayi Yu. ✉email: a.dove@bham.ac.uk; matthew.l.becker@duke.edu

Biological tissues are highly viscoelastic and dynamic[1,2]. These qualities are lacking in synthetic degradable materials that are routinely available and applied to regenerative medicine[3]. Many of the biomaterials that have been used widely for regenerative medicine, such as poly(L-lactic acid) (PLLA) and poly(ε-caprolactone) (PCL), are semi-crystalline and do not replicate the elastic properties of native tissues. These materials also exhibit anisotropic degradation as a consequence of the presence of both amorphous and crystalline domains which leads to limited control over the resorption timelines[2–4]. Attempts to engineer elastomeric materials with mechanical properties similar to native tissues have been focused on non-degradable systems[5,6]. While these have been directed towards obtaining materials that possess the elastic properties of natural rubber, they have not followed its design principles, namely the incorporation of cis-1,4 alkene segments, to control the mechanical properties. While synthetic surrogates such as cis-1,4-polyisoprene, cis-1,4-polybutadiene and analogs are available, each of these materials lack degradable units that facilitate resorption and lack the physical chemical or topological properties necessary to recapitulate a wide variety of tissues[2,7]. In addition, anionic or metallocene-based polymerization synthesis methodologies are functional group intolerant making the incorporation of bioactive groups pre- or post-polymerization that facilitate specific cellular interactions challenging.

Significant efforts have been expended to investigate degradable thermoplastic elastomers for biomaterials applications[8–23]. However, nearly all elastomer-like materials developed for tissue engineering to date require crosslinking or blending to achieve desirable mechanical and degradation properties[2]. Polyurethanes can be modified to control degradation by altering the hard segments, soft segments, and chain extenders to include varied amounts of hydrolytically degradable esters, orthoesters, amides, anhydrides, or enzymatically degradable units such as elastase-sensitive amino-acid chains[13]. The materials are known to degrade heterogeneously on account of anisotropic degradation within the soft block-forming component that leaves non-degraded hard block (typically urethane-based) components and results in exponential decreases in mechanical properties[24]. Beyond this, the resultant degradation byproducts are acidic and often elicit a strong inflammatory response[2,13,24,25]. To overcome the lack of hard block degradation, poly(ester urethane) ureas (PEUUs), which contain biodegradable urea linkages, have also been investigated[12]. These materials however largely retain bulk erosion profiles and like polyurethanes, the hard-soft block ratio dictates both the mechanical and degradative properties in a manner that cannot be decoupled[10]. Chemically crosslinked polymers like poly(glycerol sebacate) (PGS) and similar derivatives are capable of achieving elastic properties that mimic several soft tissues, and can achieve varied degradation rates by altering the crosslink density during preparation, but these materials are difficult to synthesize reproducibly, cannot be thermally processed after crosslinking, and are known to degrade too rapidly for long-term regeneration strategies (around 6 weeks in vivo)[13,17,26–28].

The need to change the chemical structure to vary the mechanical properties presents the central dogma in these materials that has made it difficult to decouple the effects of chemistry and mechanical properties on degradability and tissue regeneration. Until now, no synthetic resorbable elastomer or elastomer-like polymer system have afforded independent control of mechanical properties and degradation de novo[2]. We recently reported the first metal-free, stereocontrolled step-growth polymerization via a nucleophilic thiol-yne addition which yielded a series of thermally-processable elastomers in which the mechanical properties were controlled by the double bond stereochemistry[5,29]. The double bond stereochemistry (% cis) in

each thiol-yne step growth polymer was tuned based on solvent polarity and organic base which is able to preferentially direct the thiol addition to the cis stereochemistry. Truong et al. have shown that low and high % cis can be achieved by changing the base from Et$_3$N (pK$_a$ = 10.75) to DBU (pK$_a$ = 13.5) while maintaining the solvent (CDCl$_3$). However, moderately high % cis subunits can be achieved with Et$_3$N base when a more polar solvent such as DMSO is used. All high % cis polymers were formed using DBU/CHCl$_3$ but lower % cis contents were formed by using Et$_3$N and varying compositions of DMF and CHCl$_3$ (17:3, 7:3, and 100% DMF). However, in this initial report, the materials were non-degradable and display no significant mass loss over one year in 5 M KOH$_{(aq)}$ solution, most likely a result of resistance to ester hydrolysis due to conjugation.

In order to translate these elastomer-like systems into regenerative medicine applications, a new series of polymers have been developed that incorporates degradable succinate-based monomer units (Fig. 1A). By altering the stoichiometry of succinate incorporation, the degradation rate of the material can be tuned precisely while retaining control over the mechanical properties by maintaining the cis/trans stereochemistry of the double bond (Fig. 1B). This structural control enables the independent tuning of mechanical and degradative properties and thus overcomes a major hurdle in biomaterials. Furthermore, as a consequence of the highly hydrophobic nature of the material, they likely exhibit surface erosion behavior. In turn, these materials display excellent in vitro cell viability and have been implanted in vivo to assess degradation and the inflammatory response over 4 months in a subcutaneous rat model.

Succinic acid is found naturally within the body and can be metabolized by the Krebs cycle[30]. As such, it provided the ideal building block from which to introduce non-conjugated esters into the elastomer structure with which to influence biodegradation rates. Creation of a series of materials using a nucleophilic thiol-yne polymerization methodology was undertaken to target high cis- content at comparable molar mass ($M_w$ = 100–150 kDa) using propane-1,3-diyl dipropiolate (C$_{3A}$, 1) in combination with equimolar dithiols composed from mixtures of 1,6-hexanedithiol (C$_{6S}$) and the succinate-derived dithiol monomer bis(3-mercaptopropyl) succinate (2) (Table 1).

The ability to significantly influence mechanical properties by simply altering the cis:trans ratio by judicious choice of polymerization catalyst and solvent enables the manipulation of the materials' mechanical properties without changing the fundamental composition of the copolymer and thus affecting its degradation behavior. In order to demonstrate this, a series of materials were synthesized at constant ratio of C$_{6s}$ and succinate-based monomer, 2, (9%) while varying the cis:trans ratio between 62 and 80% which represents more than an order of magnitude change in elastic modulus of the material. The stoichiometric ratio of 2:3 and the %cis is determined easily from the splitting of the vinyl proton doublets at δ = 5.7 and 7.7 ppm (trans, 15 Hz) and δ = 5.8 and 7.1 ppm (cis, 9 Hz), respectively, in the $^1$H NMR spectra of the polymers in solution (Fig. 1B).

Uniaxial tensile testing revealed that increasing the incorporation of the succinate-based monomer 2 led to decreased ultimate tensile strength and Young's modulus and increased elongation at break (Fig. 2B, Table 2). This behavior is consistent with a more elastic material that is expected from the interruption of crystallinity through the introduction of ester groups into the main chain that disrupt chain packing and increase chain mobility. These findings were confirmed via differential scanning calorimetry (DSC) which showed that increasing %cis without altered succinate content increased the glass transition temperature in line with results from our previous work[5,29]. Significantly, the materials exhibit high thermal stability and the onset of

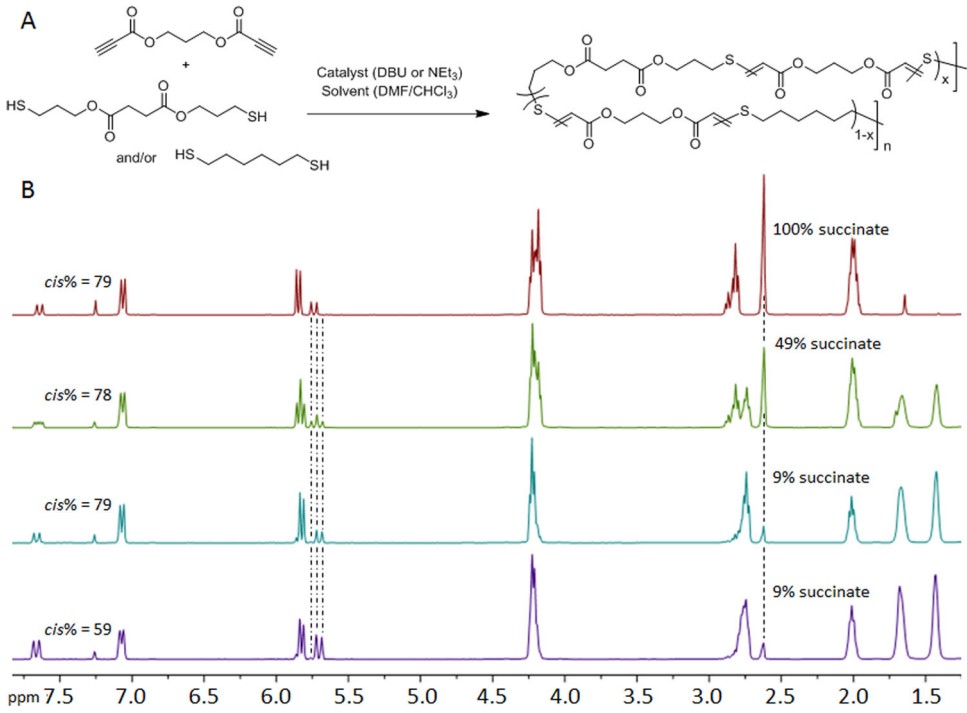

**Fig. 1 Stereocontrolled synthesis and characterization of the resorbable elastomers. A** A one-pot thiol-yne step-growth polymerization of propane-1,3-diyl dipropiolate ($C_{3A}$, **1**) with bis(3-mercaptopropyl) succinate (**2**) and with 1,6-hexane dithiol ($C_{6S}$) forms a copolymer that shows tunable degradation rates depending on the % of amount of repeat unit x (**1** + **2**) that is incorporated. **B** The stereochemistry is easily determined by the vinyl proton doublets at $\delta = 5.7$ and 7.7 ppm (*trans*, 15 Hz) and $\delta = 5.8$ and 7.1 ppm (*cis*, 9 Hz), respectively. The extent of succinate incorporation will determine both the rate and extent of degradation.

**Table 1 Polymerization conditions and characterization of all thiol-yne step growth polymers.**

| Entry | % feed of 2 | % incorporation | Monomer ratio[a] | Solvent | Base | Time (h) | % cis[b] | $M_n$ (kDa) | $M_w$ (kDa) | $Đ_M$ |
|---|---|---|---|---|---|---|---|---|---|---|
| **1** | 0 | 0 | 0.9965 | CHCl₃ | DBU | 1 | 80 | 26.4 | 147.5 | 5.60 |
| **2** | 9.0 | 9.0 | 0.9920 | CHCl₃ | DBU | 1 | 80 | 29.7 | 111.2 | 3.74 |
| **3** | 19.9 | 18.7 | 0.9922 | CHCl₃ | DBU | 1 | 79 | 35.2 | 110.8 | 3.15 |
| **4** | 50.5 | 49.4 | 0.9895 | CHCl₃ | DBU | 1 | 79 | 52.5 | 123.7 | 2.36 |
| **5** | 100 | 100 | 0.9870 | CHCl₃ | DBU | 1 | 79 | 35.9 | 112.2 | 3.12 |
| **6** | 19.8 | 19.0 | 0.9920 | DMF | Et₃N | 16 | 72 | 43.0 | 127.3 | 2.96 |
| **7** | 0 | 0 | 0.9920 | CHCl₃/DMF (7:3) | Et₃N | 16 | 62 | 37.0 | 110.8 | 3.00 |
| **8** | 11.2 | 8.7 | 0.9959 | CHCl₃/DMF (7:3) | Et₃N | 16 | 62 | 34.2 | 117.1 | 3.42 |
| **9** | 19.6 | 18.3 | 0.9920 | CHCl₃/DMF (7:3) | Et₃N | 16 | 61 | 35.3 | 107.8 | 3.05 |
| **10[c]** | 0 | 0 | 0.9950 | CHCl₃ | DBU | 1 | 80 | 24.7 | 35.4 | 1.46 |

[a]An excess of the dialkyne monomer was used to reduce any disulfide coupling and UV crosslinking side reactions.
[b]Determined by comparison of ¹H NMR integration of *cis* peaks at $\delta = 7.10$ ppm and *trans* peaks at $\delta = 7.70$ ppm.
[c]Synthesis conditions for poly(bis(4-(propioloyloxy)but-2-yn-1-yl) 3,3'-(hexane-1,6-diylbis(sulfanediyl))).

degradation temperatures exceeds 350 °C. These traits are critical for thermal processing and fabrication.

An in vitro investigation of the hydrolytic swelling and degradation behavior showed the polymers to be chemically stable with no visible degradation in PBS at ambient temperature over a 1-month period (Fig. 3). In order to accelerate the hydrolytic degradation process, the samples were incubated in 5 M KOH (aq) solution at ambient temperature. The data show that the materials with increased succinate-based monomer, **2**, yielded faster rates of degradation. Importantly, the mass loss profiles are nearly linear in nature and show no evidence of accelerated degradation as a result of acidosis and swelling via bulk erosion. The dimensions of the materials were noted to decrease concomitantly with time which is highly indicative of a

surface erosion mechanism. SEM analysis of test substrates (Fig. 3B) exposed to accelerated degradation conditions indicates uniform degradation and pitting that confirms surface erosion as the most prevalent degradation process. Taken together, these observations demonstrate that, unlike any other degradable biomaterials, the mechanical and degradation properties of these elastomer-like polymers can be controlled independently. This is a distinct difference from known polyesters. To demonstrate the potential, by careful control over double bond stereochemistry and succinate monomer (**2**) content, we prepared materials that displayed comparable degradation rates but markedly different mechanical properties and vice versa (comparable mechanical properties with markedly different degradation rates – (Fig. 2A, B). The control over each

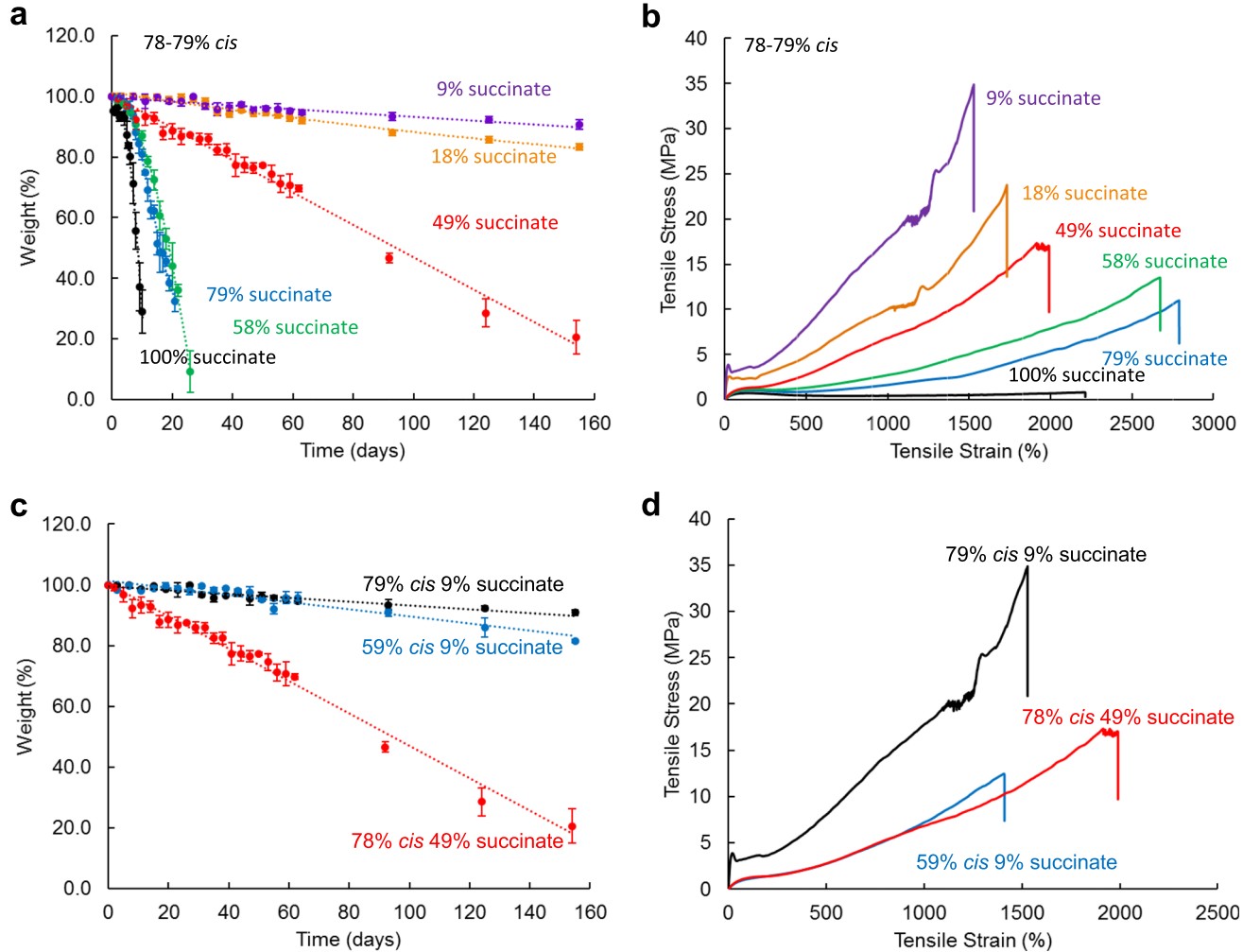

**Fig. 2 Compositional-dependent mechanical properties of the resorbable polymers. A** Mass loss as a function bis(3-mercaptopropyl) succinate (**2**) stoichiometry over time in a series of high *cis* (78–79%) elastomers show compositionally dependent linear surface erosion behavior. **B** Increasing the amount of bis(3-mercaptopropyl) succinate (**2**) which is a longer, bulkier comonomer reduced the UTS of the resulting elastomers. Mechanical properties and degradation rates are highly tunable depending on the amount of *cis*-alkene bonds in the backbone and stoichiometric control of succinate content. Succinate groups in the chemical structure provide flexibility and hydrophilicity to the polymer chains and facilitate the degradation process. The increase in chain mobility and hydrophilicity results in an increase in the number of degradable ester groups and hence the degradation rates (**C**) and a decrease of Young's modulus (**D**). Increasing the *cis*-alkene content resulted in slower degradation rates and higher Young's moduli values, decreased ultimate strain. Error bars represent one standard deviation of the mean (*n* = 3).

**Table 2 SEC, thermal and mechanical characterization of the degradable elastomers synthesized by step-growth polymerization incorporating 2.**

| % *cis*[a] | % incorporation of 2 | $M_n$ (kDa) | $M_w$ (kDa) | $Đ_M$ | $T_g$ (ºC) | $T_m$ (ºC) | $E$ (MPa) | $\varepsilon_{break}$ (%) | UTS (MPa) |
|---|---|---|---|---|---|---|---|---|---|
| 80 | 9.0 | 29.7 | 111.2 | 3.74 | 1.7 | 71.4 | 33.1 ± 3.3 | 1457 ± 312 | 34.9 ± 8.7 |
| 79 | 18.7 | 35.2 | 110.8 | 3.15 | −0.7 | 50.0 | 21.1 ± 0.7 | 1750 ± 163 | 30.1 ± 5.5 |
| 79 | 49.4 | 52.5 | 123.7 | 2.36 | −0.6 | — | 2.2 ± 0.4 | 2161 ± 158 | 18.9 ± 3.1 |
| 78 | 58.4 | 57.8 | 155.6 | 2.69 | −2.7 | — | 1.8 ± 0.4 | 3154 ± 330 | 14.6 ± 1.3 |
| 78 | 79.1 | 29.2 | 132.7 | 4.55 | −2.3 | — | 1.9 ± 0.2 | 2805 ± 149 | 10.9 ± 0.3 |
| 79 | 100 | 35.9 | 112.2 | 3.12 | −5.1 | — | 1.6 ± 0.2 | 2245 ± 1135 | 0.8 ± 0.4 |
| 62 | 8.7 | 34.2 | 117.1 | 3.42 | −8.1 | — | 2.1 ± 0.3 | 1088 ± 320 | 8.9 ± 3.5 |
| 80 | 9.0 | 29.7 | 111.2 | 3.74 | 1.7 | 71.4 | 33.1 ± 3.3 | 1457 ± 312 | 34.9 ± 8.7 |
| 79 | 49.4 | 52.5 | 123.7 | 2.36 | −0.6 | — | 2.2 ± 0.4 | 2161 ± 158 | 18.9 ± 3.1 |
| 72 | 19.0 | 43.0 | 127.3 | 2.96 | −0.2 | 49.4 | 3.2 ± 0.7 | 2158 ± 247 | 17.6 ± 3.9 |

[a]Determined by comparison of [1]H NMR spectroscopy integration of *cis* peaks at δ = 5.8 ppm and 7.1 ppm and *trans* peaks at δ = 5.7 ppm and 7.7 ppm.

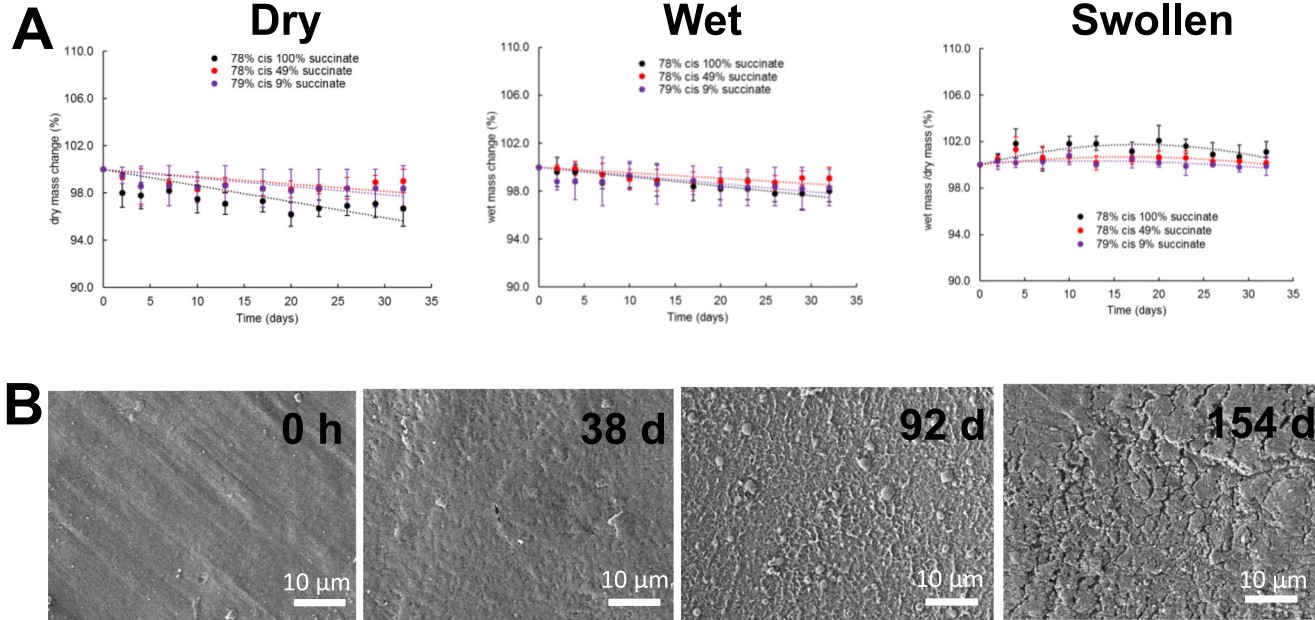

**Fig. 3 Surface Erosion and Swelling. A** Discs (4 mm diameter, 0.5 mm thick) were cut from vacuum film compression samples and placed in 1× PBS in the incubator (37 °C, 5% $CO_2$ humidified atmosphere) for up to 32 days. The data were plotted in three different ways: dry mass change compared to the original mass (degradation), wet mass change compared to the original mass (traditional swelling if no degradation), and wet mass/dry mass at each time point (swelling if degradation). The swelling behavior of the polymers was determined by tracking the wet and dry mass of the disc samples at each time point. Error bars represent one standard deviation of the mean ($n = 4$). **B** Analysis of SEM micrographs of the respective test coupons exposed to accelerated degradation conditions indicates uniform degradation and pitting indicative of surface erosion processes. Scale bars = 10 μm.

of these properties will be critical to future applications where designers will need to engineer subtle changes without returning to new synthetic methods.

To investigate the potential for use in biomaterial applications, cell viability, spreading, and proliferation assays were used as an initial method to determine the cellular responses to the elastomer-like polymers. Human mesenchymal stem cells (hMSCs) and MC3T3 cells were cultured on glass slides spin-coated with each material variant or on the control polymer, poly(L-lactide), PLLA. Cell viability was found to be higher than 95% on all samples using a Live/Dead® assay. Cell adhesion and spreading were assessed by staining F-actin, vinculin-labeled focal adhesion contacts, and cell nuclei and revealed that the hMSCs adopted an elongated and spindle-like shape on all samples (Fig. 4C). Cell proliferation was measured with a PrestoBlue® metabolic assay, after 24 h, 3 days, and 7 days of incubation. After 7 days the population of cells on each sample increased to approximately 5 times the original concentration (Fig. 4D).

One of the key aspects of a translationally relevant material is the ability to control the placement and concentration of functional species (drug, peptide, protein) on the surface of a material where the group is bioavailable to the surrounding cells and tissues[31]. While many methods are available for peptide polymer conjugation, we designed a dialkyne monomer, but-2-yne-1,4-diyl dipropiolate (**3**) that possesses an internal triple bond. As a consequence of the increased distance from the electron withdrawing groups, the reactivity of internal alkynes is distinctly different than terminal alkynes. The internal alkynes were found to be stable during the nucleophilic thiol-yne addition polymerization process with $C_{3A}$ (**1**) and $C_{6S}$ (1,6-hexanedithiol) as comonomers which left it available for selective post-polymerization functionalization of the resulting materials (Fig. 4E). Following the polymerization and a film casting process, a Megastokes®-673-azide dye surrogate (Fig. 4F) was covalently tethered to the internal alkyne functionalized poly(bis(4-(propioloyloxy)but-2-yn-1-yl)-3,3'-(hexane-1,6-

diylbis(sulfanediyl))) using Cp*RuCl(COD) as a catalyst[32]. After washing, the film remained fluorescent thus evidencing the conjugation of the dye to the polymer. To extend the concept and show utility from a biomaterials viewpoint, we also used this methodology to attach an azide-functionalized GRGDS peptide to the films. While only an initial demonstration, the presence of the adhesion peptide had a distinct influence on the cell adhesion and spreading properties (Figs. 4G and 4H). Following peptide conjugation, increased cell adhesion, spreading and integrin-associated actin fiber formation was evident in the RGD derivatized films relative to the unfunctionalized films. Future studies are developing this technique to apply other bioactive groups designed to influence specific cellular activities.

The lack of cytotoxicity and enhanced cellular activity confirmed that the thiol-yne stereoelastomer materials could be implanted in vivo for tissue compatibility studies. Elastomeric discs possessing various cis content and succinate stoichiometry were implanted subcutaneously for 4 months to observe the degradation behavior and tissue inflammatory responses in vivo. Significantly, no gross inflammation, which would appear as a dense calcified capsule, was evident from macroscopic images of the samples taken at each timepoint (Fig. 5B). Sections stained with hematoxylin and eosin (H&E) were analyzed for inflammatory responses in the form of fibrous capsule formation. Sections stained with H&E were also assessed for inflammatory cell infiltration. Fibrous encapsulation occurred as expected, and the granuloma grew thicker over 4 months of incubation with no significant difference compared to PLLA (Fig. 5). The granuloma was less than 200 μm thick for all samples, which has previously been reported as acceptable in terms of tissue compatibility for long-term implants[4,33,34]. This is indicative of the tunable degradation profiles from varied succinate content as increasing implant degradation rates correlates with greater cellular remodeling processes[34].

H&E slides were quantitatively analyzed for neutrophils, lymphocytes, plasma cells, single macrophages, multinucleated giant

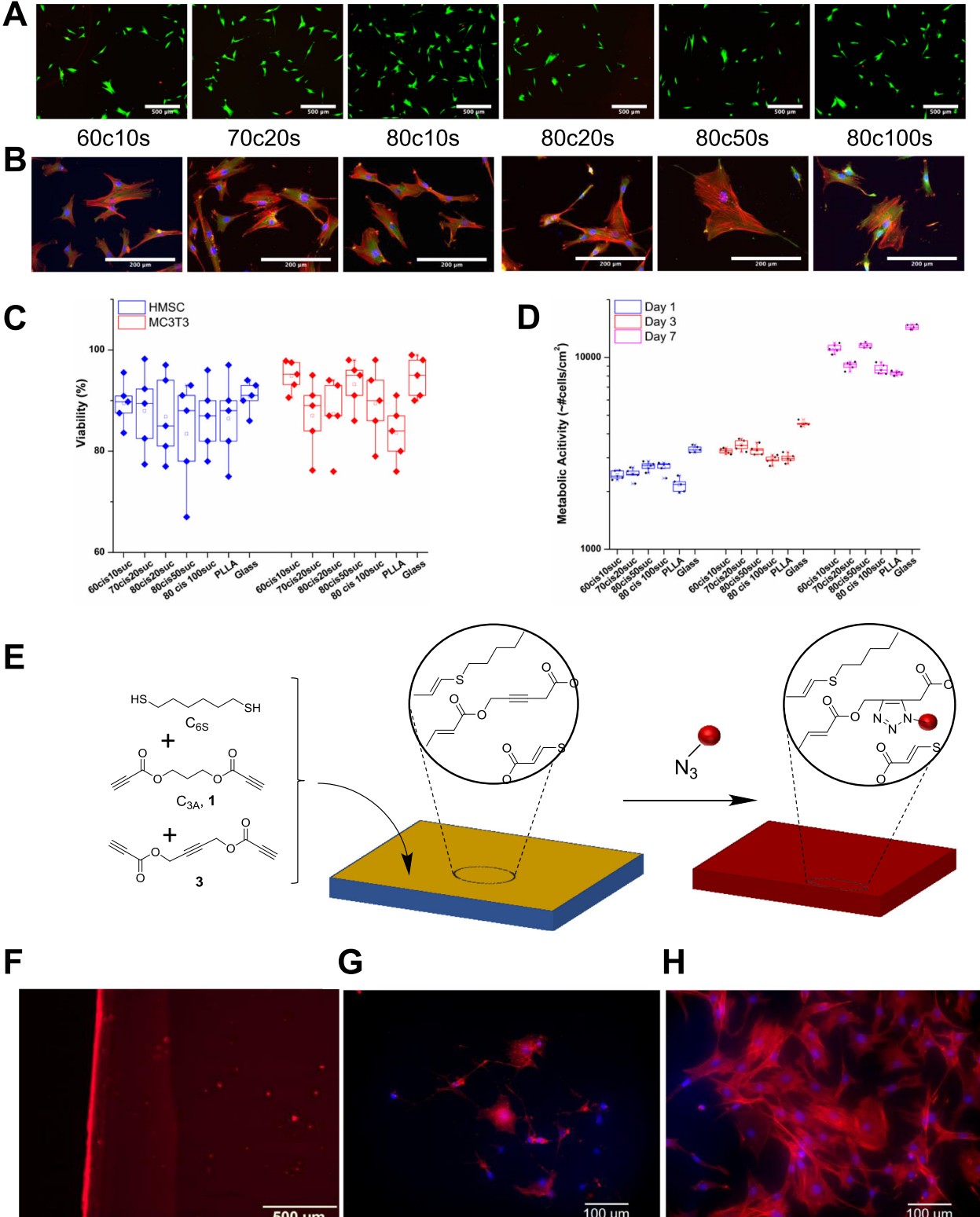

**Fig. 4 Cell Viability and Post polymerization Functionalization. A** Live/Dead® staining of cells incubated on each substrate for 24 h. Calcein-AM was used to stain live cells (green) and ethidium homodimer-1 was used for dead cells (red). Scale Bars are 200 μm. **B** Representative fluorescence pictures of *h*MSCs cultured on degradable polymer substrates for 72 h. Scale bars are 500 μm. **C** Quantitative cell viability data showed >95% viability after 24 h in both hMSCs and MC3T3 fibroblasts. Error bars represent one standard deviation of the mean ($n = 5$). **D** Cell metabolic activity showed an increase in approximate cell number on degradable polymer substrates over 7 days. Error bars represent one standard deviation of the mean ($n = 5$). **E** To demonstrate the ability to derivatize the polymers post polymerization, a Megastokes®-673-azide dye surrogate specifically binds to the internal alkyne functionalized polymer (**F**). Scale bar is 500 μm. Cell spreading on (poly(bis(4-(propioloyloxy)but-2-yn-1-yl)-3,3'-(hexane-1,6-diylbis(sulfanediyl)))) films without (**G**) and with (**H**) RGD functionalization. Increased cell adhesion, spreading and integrin-associated actin fiber formation was evident in the RGD derivatized films, indicating that RGD conjugation was successful. Scale bars are 100 μm.

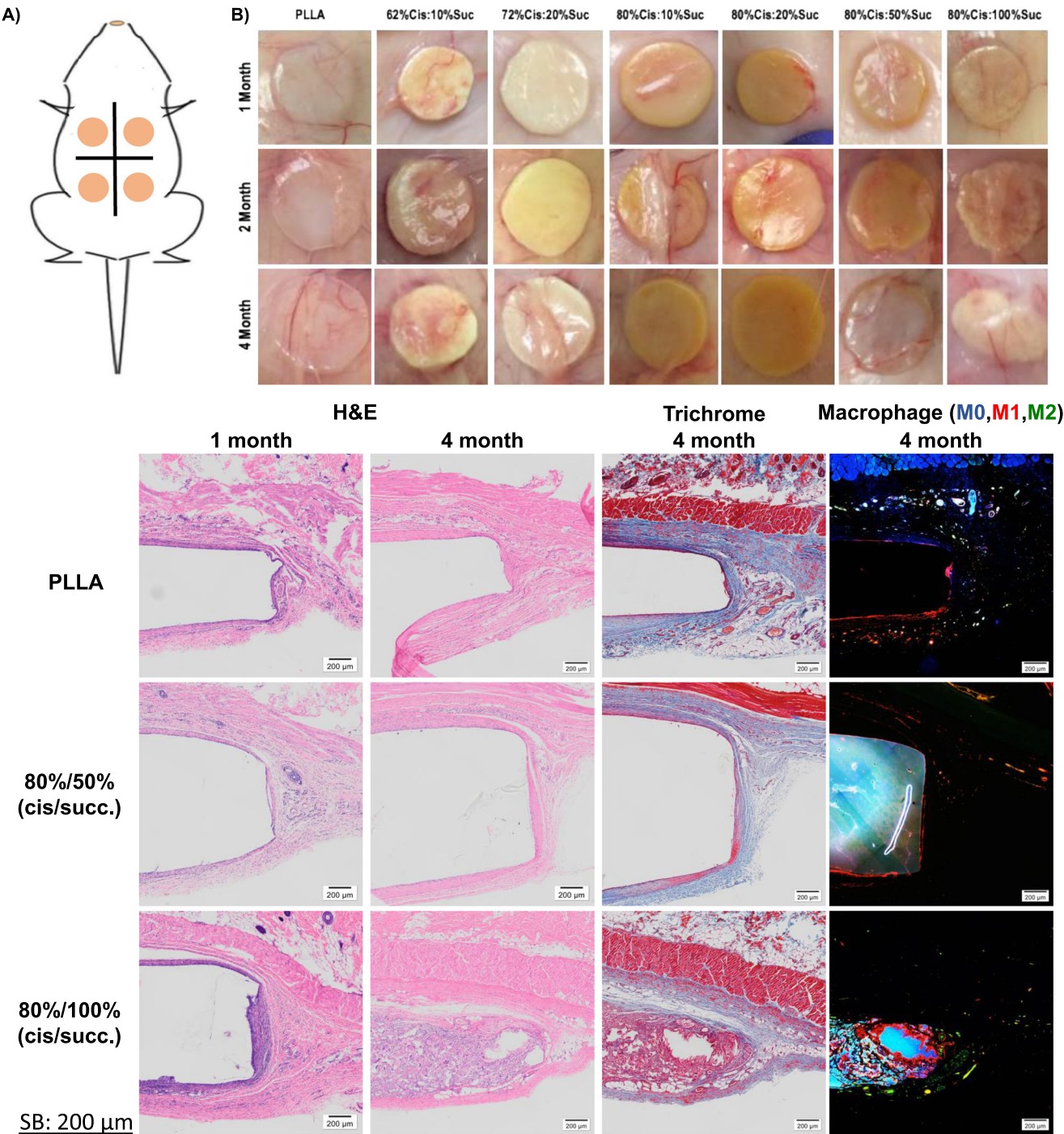

**Fig. 5 Subcutaneous in vivo degradation of Poly(L-lactic acid) (PLLA), 80% *cis*/50% succinate and 80% *cis*/100% succinate over a 4-month timeframe.** Surgical procedures with subcutaneous implantation involved a small incision, polymer disc insertion, and incision closure with Michel-clips. Four samples were implanted per animal (**A**). (**B**) Following extraction, the implants can be visualized in the host tissue using Hematoxylin and Eosin (H&E) and Masson's Trichrome staining. As seen, there are almost no macroscopic indications of an inflammatory response. Whole-mount cross-section images showing thick fibrous encapsulation surrounding PLLA after 4 months of incubation in vivo are observed. Similar behavior to PLLA is observed for 80% cis/ 50% succinate at 1- and 4-months implantation. Alternatively, the early stages of cellular infiltration are noticed in 80% *cis*/100% succinate after only 1 month (I). After 2- (Supplementary Fig. 23) and 4- months, noticeable shrinking/resorption of the polymer was seen with continued cellular infiltration. Degradation after 4 months is nearly complete with cells, deposited collagen, and tissue fully encompassing the polymer area. Blood vessel sprouts and multinucleated giant cells are noticeable throughout the polymer space that has been resorbed. Trichrome images show collagen deposition and immunohistochemistry staining macrophages for pro-inflammatory (**M1**), non-activated (**M0**), and anti-inflammatory (**M2**) macrophages show degradation induced inflammation and remodeling. Inset scale bar = 200 μm. Shown micrographs are representative of histology specimens ($n = 4$) from each of six independent implants ($n = 6$) for each material.

cells, and necrosis at the 1, 2, and 4-month time points following a subcutaneous rat implant model. Additionally, each slide was assessed for inflammatory cell infiltrate based on a modified scoring system outlined by the International Organization for Standardization (ISO 10993-6 Annex E) by a board-certified veterinary pathologist. The numbers of inflammatory cells were estimated in a 400× field using light microscopy images, and a score was assigned for each inflammatory cell type as denoted in Table 3. The most severely affected region of the evaluated tissue was utilized to assign a score. The severity of necrosis was judged by the percentage of the fibrous capsule exhibiting evidence of necrosis (pyknosis, karyorrhexis, or karyolysis) not including any inflammatory cell infiltrate. Comparisons were made between elastomers with the 80%*cis* content and containing a range of % succinate content (Fig. 2). A striking difference was apparent between elastomers with lower succinate content from 10 to 50% compared to the 100% succinate-containing elastomer after only one month, and increasingly over 4 months (Table 2).

Full tissue infiltration into the polymer space occurred as the 80% *cis*/100% succinate polymer degraded (Fig. 5). This was noted by an increase in the total number of inflammatory cells into the capsule space with a total score of 8.6 ± 1.5, 9.1 ± 1.2, and 10.9 ± 1.9 for 1, 2, and 4-months respectively. This was noticeably greater than all other materials which elicited total scores ranging between 4.3 ± 1.2–4.9 ± 1.8, 3.8 ± 1.4 - 4.7 ± 1.7, and 2.6 ± 1.3–4.0 ± 2.0 for 1, 2, and 4-months respectively. For reference, medical-grade polypropylene has scored around 7.5 in a similar recent study[34]. While used widely in the clinic, this value would be noted as a persistent low-level inflammatory response. In each of the materials above, the reported values are 30-50% less than polypropylene. The investigation of these materials in more translationally relevant applications is ongoing.

Very few multinucleated giant cells were found surrounding the implants and there was no evidence of necrosis, even at extended timepoints. The only samples with a few multinucleated giant cells were the 80% *cis*/50% succinate and 80% *cis*/100% succinate elastomers, where giant cells were found infiltrating degraded polymer areas. Multinucleated giant cells attempt to encapsulate portions of the foreign body that have broken away as well as releasing factors that degrade extracellular matrix and cause damage and degradation of the implanted material, and thus are regarded as an obstacle for clinical translation of biomaterials[35,36]. The absence of multinucleated giant cells shows a limited foreign body response over the period of the experiment. Macrophage staining of 4 month 80c100s polymer shows evidence of a robust inflammatory response that is expected to occur during the degradation, resorption, and remodeling process. CD68 was used as a pan-macrophage (M0, blue) marker, CCR7 was used to indicate classically activated macrophages (M1, red), and CD206 was used to indicate the presence of alternatively activated macrophages (M2, green). Non-specific control staining shows subtle autofluorescence of the stereoelastomers inhibits quantitative analysis of macrophage presence. The presence of M2 macrophages indicates that a transition to a remodeling phase is likely occurring[37]. Trichrome staining shows no evidence of capsule formation in the stereoelastomer samples while a thicker layer of collagen deposition surrounds the PLLA implant. The semicrystalline PLLA control material in this study is likely not degrading quickly compared to previous literature reports of amorphous PDLLA where multinucleated giant cell numbers were extremely high[37,38]. The surface chemistry differences imparted by the crystalline domains of the materials play an important role in the amount and conformation of protein absorption, and this subsequently affects the process of multinucleated giant cell formation.

**Table 3 Modified 10993-6 inflammatory histological response analysis.**

| Cell type response | PLLA | | | 62/10 | | | 80/10 | | | 80/50 | | | 80/100 | | |
|---|---|---|---|---|---|---|---|---|---|---|---|---|---|---|---|
| | 1 Month | 2 Month | 4 Month | 1 Month | 2 Month | 4 Month | 1 Month | 2 Month | 4 Month | 1 Month | 2 Month | 4 Month | 1 Month | 2 Month | 4 Month |
| Neutrophils | 0.3 ± 0.5 | 0.6 ± 0.9 | 0.2 ± 0.4 | 0.1 ± 0.3 | 0.6 ± 0.7 | 0.2 ± 0.4 | 0.1 ± 0.2 | 0.7 ± 0.9 | 0.1 ± 0.2 | 0.4 ± 0.5 | 0.9 ± 1.0 | 0.3 ± 0.5 | 0.4 ± 0.5 | 1.1 ± 0.6 | 0.1 ± 0.2 |
| Lymphocytes | 2.5 ± 0.6 | 2.0 ± 0.7 | 2.2 ± 0.7 | 2.8 ± 0.5 | 2.2 ± 0.7 | 2.2 ± 0.8 | 2.7 ± 0.5 | 2.4 ± 0.8 | 2.0 ± 0.8 | 3.7 ± 0.4 | 1.8 ± 0.8 | 1.9 ± 0.8 | 3.7 ± 0.4 | 2.8 ± 0.7 | 3.2 ± 0.7 |
| Plasma cells | 0.1 ± 0.2 | 0.1 ± 0.3 | 0 ± 0 | 0.1 ± 0.2 | 0.1 ± 0.2 | 0 ± 0 | 0 ± 0 | 0 ± 0 | 0.1 ± 0.2 | 0.6 ± 0.6 | 0.3 ± 0.4 | 0 ± 0 | 0.6 ± 0.6 | 0 ± 0 | 1.4 ± 0.6 |
| Single macrophages | 1.9 ± 1.2 | 0 ± 0 | 1.5 ± 1.4 | 1.6 ± 1.3 | 1.0 ± 0.9 | 1.1 ± 1.2 | 1.6 ± 1.1 | 1.2 ± 0.9 | 0.5 ± 0.8 | 2.9 ± 0.2 | 1.7 ± 1.1 | 0.5 ± 0.9 | 2.9 ± 0.2 | 1.4 ± 1.0 | 3.8 ± 0.4 |
| Multinucleated giant cells | 0 ± 0 | 0 ± 0 | 0 ± 0 | 0 ± 0 | 0 ± 0 | 0 ± 0 | 0 ± 0 | 0 ± 0 | 0 ± 0 | 0.9 ± 0.5 | 0.1 ± 0.2 | 0 ± 0 | 0.9 ± 0.5 | 0.1 ± 0.2 | 2.5 ± 0.9 |
| Necrosis | 0 ± 0 | 0 ± 0 | 0 ± 0 | 0 ± 0 | 0 ± 0 | 0 ± 0 | 0 ± 0 | 0 ± 0 | 0 ± 0 | 0 ± 0 | 0 ± 0 | 0 ± 0 | 0 ± 0 | 0 ± 0 | 0 ± 0 |
| | 4.8 ± 1.6 | 4.0 ± 1.6 | 4.0 ± 2.0 | 4.6 ± 1.7 | 3.8 ± 1.4 | 3.5 ± 1.6 | 4.3 ± 1.2 | 4.2 ± 2.0 | 2.6 ± 1.3 | 4.9 ± 1.8 | 4.7 ± 1.7 | 2.7 ± 1.2 | 8.6 ± 1.5 | 9.1 ± 1.2 | 10.9 ± 1.9 |

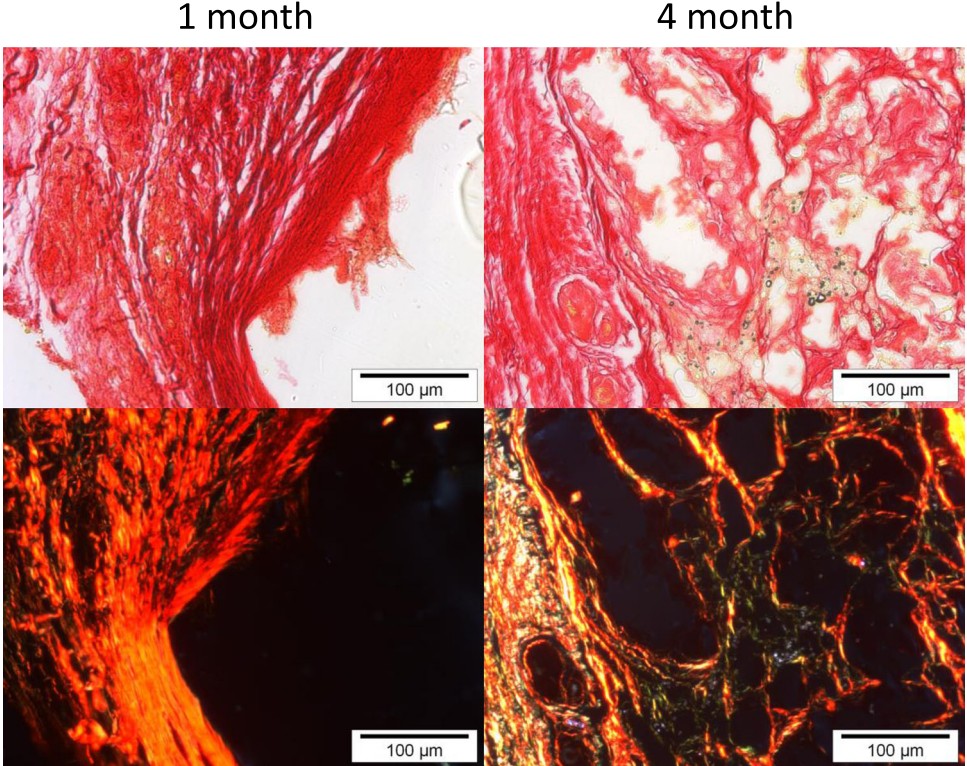

**Fig. 6 Picrosirius Red staining of 80*cis*:100 succinate elastomers after 1 month and 4 months of subcutaneous incubation.** Collagen deposition and maturation occurred throughout the polymer space with different orientations representing both mature and developing collagen through the center of the polymer area. Scale bars are 100 µm. Shown micrographs are representative of histology specimens ($n = 4$) from each of six independent implants ($n = 6$) for each material.

Picrosirius red (PSR) staining is a commonly used histological technique to visualize collagen in paraffin-embedded tissue sections[39]. PSR stained collagen appears red in optical microscopy. However, it is largely unknown that PSR stained collagen also shows a red fluorescence, whereas live cells have a distinct green autofluorescence. As shown in Fig. 6, Picrosirius Red staining is present at the edge of the degrading stereoelastomers (1 month) and throughout the site formerly occupied by the degraded 80*cis*:100 succinate materials after 4 months of subcutaneous incubation. This clearly shows that collagen deposition and maturation occurred throughout the space formerly occupied by the polymer.

These results, along with a decrease in size of the 80% *cis*/100% succinate, shows that tissue remodeling and near complete polymer resorption occurs over a 4-month time frame (Fig. 5). Remodeling of the degraded polymer over 4 months is comparable to that seen in poly(glycerol succinate) materials after 9 weeks of implantation, suggesting that this material is suitable for tissue regeneration, particularly for situations where the polymer must endure beyond 2 months to provide appropriate mechanical load reinforcement[40]. Future studies for these materials will observe the effects of degradation and associated inflammation for the slower degrading variants. Many recent findings have noted the importance of fine-tuning degradation rates to match regeneration rates for optimal tissue growth and mechanical reinforcement as needed throughout healing[2,24,29,41]. Significantly, this study has shown significant tissue growth into non-functionalized bulk implants where inflammation and granuloma formation was limited. The versatility in the synthesis of these elastomer-like polymers with controlled variability of mechanical properties and degradation rates marks significant progress in the field of degradable biomaterials.

In summary, we have developed a series of highly tunable and resorbable elastomer-like polymers that afford concomitant control of mechanical and degradation properties. These materials have shown excellent cellular responses in vitro and possess limited inflammatory responses in vivo. Most importantly, the variants containing 100% succinate incorporation were capable of degrading in vivo over a period of four months and were replaced with mature and developing tissues. These responses show that these materials are non-toxic and further, will provide a new developmental platform for regeneration of tissues with varied mechanical and degradation requirements. Future studies on these materials will include optimization of material properties, control/inhibition of crosslinking, post-polymerization functionalization with bioactive species and assessment of mechanical properties throughout degradation.

## Methods

**Materials.** The following chemicals were used as received: acetone (Sigma-Aldrich, ≥99.0%), chloroform (CHCl₃: VWR Chemicals, 99%), d-chloroform (CDCl₃: Apollo, >99%), 1,8-diazabicyclo[5.4.0]undec-7-ene (DBU: Sigma-Aldrich, 98%), diethyl ether (Et₂O: Sigma-Aldrich, ≥99.8%), *N,N* dimethylformamide (DMF: Fisher Scientific, LR grade), 2,6-di-*tert*-butyl-4-methylphenol (BHT: Alfa Aesar, 99%), ethyl acetate (EtOAc: Fisher Scientific, LR grade), hexane (Hex: VWR Chemicals, 99%), magnesium sulfate (MgSO₄: anhydrous, Fisher Scientific, LR grade), 3-mercapto-1-propanol (Tokyo Chemical Industry Ltd. UK, 96%), 1,3-propanediol (Sigma-Aldrich, 98%), propiolic acid (Acros Organics, 98%), silica gel (SiO₂: Apollo Scientific, 40-63 micron), sodium chloride (NaCl: Fisher Scientific, > 99%), sodium hydrogen carbonate (NaHCO₃: Fisher Scientific, >99%), sulfuric acid (Fisher Scientific, >95%), triethylamine (Et₃N: Fisher Scientific, LR grade). 1,6-hexanedithiol (Sigma-Aldrich, ≥97%) was vacuum distilled prior to use and stored in Young's tapped ampuoles under N₂. Poly(ʟ-lactic acid) (PLLA) (Ingeo™ Biopolymer 3100HP) was ordered from NatureWorks.

**Molecular structure and thermal characterization.** Size exclusion chromatography (SEC) analyses were performed on a system composed of a Varian 390-LC-

Multi detector using a Varian Polymer Laboratories guard column (PLGel 5 μM, 50 × 7.5 mm), two mixed D Varian Polymer Laboratories columns (PLGel 5 μM, 300 × 7.5 mm) and a PLAST RT autosampler. Detection was conducted using a differential refractive index (RI) detector. The analyses were performed in CHCl₃ at 40 °C and containing 0.5% w/w Et₃N at a flow rate of 1.0 mL/min. Linear polystyrene (PS) (162–2.4 × 10⁵ g mol⁻¹) standards were used to calibrate the system. EcoSEC HLC-8320 GPC (Tosoh Bioscience LLC, King of Prussia, PA) equipped with a TSKgel GMH$_{HR}$-M mixed bed column and refractive index (RI) detector was performed to analyze poly(bis(4-(propioloyloxy)but-2-yn-1-yl)-3,3'-(hexane-1,6-diylbis(sulfanediyl))) (**10**). Molecular mass was calculated using a calibration curve determined from polystyrene standards (PStQuick MP-M standards, Tosoh Bioscience, LLC) with DMF with 0.1 M LiBr as eluent flowing at 1.0 mL min⁻¹ at 323 K, and a sample concentration of 3 mg mL⁻¹.

Nuclear magnetic resonance (¹H, ¹³C) spectra were recorded in CDCl₃ on a Bruker DPX-400 spectrometer at 298 K. Chemical shifts are reported as δ in parts per million (ppm) and referenced to the chemical shift of the residual solvent resonances (CDCl₃ ¹H: δ = 7.26 ppm, ¹³C: δ = 77.16 ppm). The resonance multiplicities are described as s (singlet), d (doublet), t (triplet), q (quartet) or m (multiplet).

Thermogravimetric analysis (TGA) (Q500, TA Instruments, New Castle, DE) was performed over a temperature range from 0 to 600 °C at a heating rate of 10 °C/min. A 5% loss in mass was used to determine the onset temperature of degradation ($T_d$).

Differential scanning calorimetry (DSC) (Q2000, TA Instruments, New Castle, DE) was used with a temperature range from −20 to 200 °C and a scanning rate of 10 °C/min in a heating/cooling/heating mode to determine glass transition temperatures ($T_g$) of polymers obtained during the second heating cycle.

### Monomer synthesis

*Propane-1,3-diyl dipropiolate ($C_{3A}$, 1)*. 1,3-propanediol (20.00 g, 0.263 mol) was added to a 1 L single neck round bottom flask. To this was added toluene (100 mL) and benzene (100 mL). Two drops of H₂SO₄ were added and the solution was allowed to stir at room temperature for 5 min before adding propiolic acid (50.00 g, 0.714 mol). A Dean-Stark apparatus with condenser was fitted and the reaction was then refluxed for 16 h at 120 °C or until the required amount of water was collected. The solution was then cooled to room temperature and solvent-extracted with saturated NaHCO₃ solution (2 × 200 mL) to remove any residual acids. The organic phase was then collected, dried over MgSO₄, filtered, and reduced in volume to dryness. The product was purified on silica gel isocratically using 4:1 hexane/EtOAc and collecting the 1st fraction. After removal of the solvent, the final product was further purified by distillation under high vacuum at 160 °C to yield colorless oil that slightly crystallized on sitting (24.63 g, 52% yield). $R_f$ (3:2 Hex/EtOAc) = 0.43; Melting point: 25 °C; ¹H NMR (500 MHz, CDCl₃) δ 4.30 (t, $^3J_{HH}$ = 6.2 Hz, 4H), 2.88 (s, 2H), 2.19–1.96 (m, 2H); ¹³C NMR (125 MHz, CDCl₃) δ 152.6, 75.3, 74.5, 62.6, 27.5; ESI-MS Calcd for C₉H₈O₄Na (M + Na): 203.0, Found: 203.0; Anal Calcd for C₉H₈O₄: C 60.00; H 4.48%. Found: C 59.70; H 4.41%.

*Bis(3-mercaptopropyl) succinate (2)*. 3-mercaptopropanol (7.30 g, 0.079 mol) was added to a 250 mL single neck round bottom flask. To this was added toluene (60 mL) and benzene (60 mL). Two drops of H₂SO₄ were added and the solution was allowed to stir at room temperature for 5 min before adding succinic acid (4.40 g, 0.037 mol). A Dean-Stark apparatus with condenser was fitted and the reaction was then refluxed for 16 h at 120 °C or until the required amount of water was collected. The solution was then cooled to room temperature and solvent removed by vacuum transfer. The product was resolubilized in CHCl₃ (100 mL) and extracted with saturated NaHCO₃ solution (2 × 200 mL) to remove any residual acids. The organic phase was then collected, dried over MgSO₄, filtered, and reduced in volume to dryness. The product was then purified on silica gel isocratically using 3:2 hexane/EtOAc and collecting the 1st fraction. After removal of the solvent, the final product was further purified by distillation under high vacuum (0.15 Torr) at 220 °C to yield colorless oil (7.8 g, 79% yield). $R_f$ (3:2 Hex/EtOAc) = 0.4; ¹H NMR (400 MHz, CDCl₃) δ 4.21 (t, $^3J_{HH}$ = 6.2 Hz, 4H), 2.62 (s, 4H), 2.58 (q, $^3J_{HH}$ = 6.6 Hz, 4H), 1.40 (t, $^3J_{HH}$ = 8.1 Hz, 2H); ¹³C NMR (100 MHz, CDCl₃) δ 172.3, 62.9, 32.9, 29.2, 21.2; ESI-MS Calcd for C₁₀H₁₈O₄S₂Na⁺ (M + Na⁺): 289.1, Found: 289.0; Anal Calcd for C₁₀H₁₈O₄S₂: C 45.09; H 6.81%. Found: C 59.70; H 4.41%.

*Sodium propiolate*. Sodium propiolate was synthesized according to the procedure described by Bonnesen et al. [1], Sodium hydroxide (0.645 g, 0.016 mol) was dissolved in methanol (50 mL) in a 250 mL round-bottom flask and protected from light. The solution was cooled to 0 °C for 10 min. Then propionic acid (1.00 mL, 0.016 mol) was added with stirring. The solution was allowed to warm to ambient temperature and stirred for additional 2 h. The solvent was then removed by rotary evaporation. A white solid product was formed and dried under high vacuum to yield **3** (1.44 g, 97%). The product should be stored in the dark due to light sensitivities. ¹H NMR (300 MHz, CD₃OD) δ 2.95 (s, 1H). ¹³C NMR (75 MHz, CD₃OD) δ 160.64, 81.83, 69.12.

*But-2-yne-1,4-diyl bis(4-methylbenzenesulfonate)*. But-2-yne-1,4-diyl bis(4-methylbenzene sulfonate) was synthesized according to the procedure described by Maisonial et al.[2] Briefly, *p*-toluenesulfonyl chloride (24.00 g, 0.126 mol) and

1,4-butynediol (4.00 g, 0.046 mol) were dissolved in Et₂O (300 mL). The mixture was cooled to −15 °C for 15 min before potassium hydroxide (16.00 g, 0.285 mol) was added slowly. The resulting solution was stirred at 0 °C for 3 h and poured into ice water (300 mL). When the solution reached ambient temperature, the solution was extracted with DCM (200 mL × 3) and the organic layer was collected, dried with anhydrous Na₂SO₄, filtered, and concentrated. The solid was washed with Et₂O (100 mL × 3) and dried under vacuum 24 h to yield **4** (14.66 g, 80%) as a white solid. ¹H NMR (300 MHz, CDCl₃) δ 7.77 (d, J = 8.8 Hz, 4H), 7.34 (d, J = 8.8 Hz, 4H), 4.58 (s, 4H), 2.46 (s, 6H). ¹³C NMR (75 MHz, CDCl₃) δ 145.54, 132.80, 130.01 (×2), 128.16 (×2), 81.04, 57.21, 21.76.

*But-2-yne-1,4-diyl dipropiolate (3)*. Under reduced light conditions, sodium propiolate (7.600 g, 0.083 mol) and But-2-yne-1,4-diyl bis(4-methylbenzene sulfonate) (12.00 g, 0.030 mol) were dissolved in DMF (120 mL), and the mixture was heated to 50 °C, and allowed to stir for 24 h. After the reaction was cooled down to ambient temperature, a saturated solution of NH₄Cl (200 mL) was added to the mixture and the reaction was stirred for 10 min. The mixture was extracted with DCM (150 mL × 3) and the organic extracts were combined, extracted with saturated solution of NaHCO₃ (150 mL × 3). The organic layer was combined and dried over anhydrous Na₂SO₄, filtered, and concentrated. The residue was purified by flash column chromatography on silica gel (EtOAc/hexanes 1:3; $R_f$ = 0.30). After removal of the solvent, the final product was further purified by distillation under high vacuum at 110 °C to yield a colorless oil (3.76 g, 65%). ¹H NMR (300 MHz, CDCl₃) δ 4.81 (s, 4H), 2.97 (s, 2H). ¹³C NMR (75 MHz, CDCl₃) δ 151.78, 80.60, 76.34, 73.82, 53.28. ESI-MS for C₁₀H₆O₄Na, m/z theoretical: [M + Na]⁺ = 213.02 Da, observed: [M + Na]⁺ = 213.0 Da.

**General procedure for thiol-yne step growth polymerization.** An example of the thiol-yne step growth polymerization is as follows: 1,6-hexanedithiol (0.73 g, 4.9 × 10⁻³ mol) and **2** (0.32 g, 1.2 × 10⁻³ mol) were added to a 20 mL scintillation vial. Propane-1,3-diyl dipropiolate (1.10 g, 6.1 × 10⁻³ mol) was added to the solution by quantitative transfer with CHCl₃ (12 mL). The solution was then cooled to −15 °C with stirring for 15 min before DBU (9 μL, 6.0 × 10⁻⁵ mol) was added. The addition of DBU produced an exotherm causing the solvent to bubble. After 2 min of stirring, the reaction was then allowed to warm to room temperature and continued to stir, during which time the solution became very viscous. After 1 h, the solution was diluted with CHCl₃ (8 mL). The polymer solution was then precipitated into 1:1 diethyl ether/acetone (200 mL) and collected by decanting the supernatant. The polymer was then redissolved in CHCl₃ (20 mL) and reprecipitated into 1:1 diethyl ether/acetone (200 mL). The polymer was again redissolved in CHCl₃ (20 mL) and 100 mg BHT (5 %w/w) was added. The final solution was then precipitated into n-hexane (200 mL), collected by decanting the supernatant, and dried *in vacuo* at room temperature for 24 h. SEC (CHCl₃ + 0.5% Et₃N) $M_n$ = 35.2 kDa, $M_w$ = 110.8 kDa, $M_p$ = 106.9 kDa, $Đ_M$ = 3.15. ¹H NMR (CDCl₃, 400 MHz) % incorporation of **2** = 18.7%; % *cis* = 79%.

**Variation of molecular mass.** The molecular mass of the thiol-yne step growth polymers was varied by changing the amount of dithiol in relation to the dialkyne such that the dialkyne was always in excess. Monomer ratios were determined using the extended Carothers equation for one monomer in excess (assuming $p \rightarrow 100\%$)[1].

**Procedure of thiol-yne step-growth polymerization for but-2-yne-1,4-diyl dipropiolate.** 1,6-Hexanedithiol (4.300 g, 0.028 mol) was added into 500 mL round bottom flask and but-2-yne-1,4-diyl dipropiolate (**5**) (5.500 g, 0.029 mol) was added to a 500 mL round bottom flask with 200 mL CHCl₃. The solution was then cooled to −15 °C with stirring for 20 min before DBU (44 μL, 29 mmol) was added in one portion. Notably, the addition of DBU caused the solvent to bubble due to an exothermic reaction. After stirring for 10 min, the reaction was allowed to warm to room temperature and continued to stir. After 1 h, a couple of drops of but-2-yne-1,4-diyl dipropiolate **5** in CHCl₃ (5 mL) was added into the reaction solution. After stirring for an additional 30 min, the solution was diluted with CHCl₃ (50 mL) and BHT (0.48 g, 0.002 mol) before the precipitation steps. The polymer solution was then precipitated into diethyl ether (1.5 L) and collected by decanting the supernatant. The polymer was then redissolved in CHCl₃ (150 mL) and reprecipitated into diethyl ether (1.5 L), collected by decanting the supernatant, and dried by high vacuum system at room temperature for 24 h to give pale yellow polymer poly(bis(4-(propioloyloxy)but-2-yn-1-yl)-3,3'-(hexane-1,6-diylbis (sulfanediyl))) (**10**) (8.3 g, 85%). SEC (DMF + 0.1 M LiBr, based on PS standards) $M_n$ = 24.7 kDa, $M_w$ = 35.4 kDa, $Đ_M$ = 1.46. ¹H NMR (CDCl₃, 300 MHz) % *cis*: % *trans* = 78 %: 22 %. DSC: $T_g$ = 22 °C. TGA: $T_d$ = 287 °C.

**Post-polymerization functionalization with GRGDS peptide.** The end-capped polymer poly(bis(4-(propioloyloxy)but-2-yn-1-yl) 3,3'-(hexane-1,6-diylbis(sulfanediyl))) (**10**) (300 mg; $M_n$ = 24.7 kDa, $M_w$ = 35.4 kDa, $Đ_M$ = 1.46.) and Cp*RuCl (COD) (1 mg, 0.003 mmol) were added to a 100 mL two neck round bottom flask and the round bottom flask was evacuated and purged with N₂ three times before dried DMF (40 mL) was added. Then, 3 wt% N₃-GRGDS peptide (FW = 629.63 g/mol; 9 mg) was dissolved in dried DMF (5 mL) and added into the reaction

solution by syringe and allowed to stir for 12 h. The solution of GRGDS peptide functionalized polymer was precipitated into ethanol (500 mL), collected and dried under vacuum for 24 h to afford GRGDS peptide functionalized polymer poly(bis (4-(propioloyloxy)but-2-yn-1-yl)-3,3′-(hexane-1,6-diylbis(sulfanediyl)))-GRGDS.

**Mechanical testing**. Destructive tensile tests were performed to determine the effects of altered *cis*-alkene and succinate incorporation on Young's Modulus ($E$) and ultimate strain ($\varepsilon_U$). Samples ($n = 3$) were prepared using vacuum film compression (Technical Machine Products, Cleveland, OH) to press films measuring 50 mm × 50 mm × 0.5 mm. Polymer was preheated at 120 °C for 15 min, and then compressed at 10,000 lbs of pressure for 4 min before cooling rapidly under vacuum. Tensile bars were cut using a custom-made dog bone-shaped die cutter and were pulled at several rates to determine the rate at which equilibrium modulus of all samples could be obtained. Rates tested were 0.1, 1, 5, 10, and 20 mm/min. A rate of 10 mm/min was determined to be appropriate. Samples were tested in an Instron® (5567) equipped with a 100 N load cell. The results were recorded using Bluehill® 3 software (Instron®, Norwood, MA). The modulus values quoted are from the tangent of the initial yield point at low strain where it exists or over the 2–10% strain regime. The results are the average values of 5 ($n = 5$) individual measurements for each material.

**Accelerated in vitro degradation studies**. A film in 0.5 mm thickness of each elastomer was prepared from vacuum film compression using the same method as stated above. Discs with 4 mm in diameter were cut from the film and placed in 5 M NaOH solution in the incubator (37 °C, 5% $CO_2$ humidified atmosphere) for up to 200 days. The films absorbed, swell, degraded and the 5 M NaOH solution was changed every week to ensure the degradation process. At specified intervals, the samples were removed, dried, and weighed. The results of mass changes are the average values of four ($n = 4$) individual samples for each material at each time point.

**Biological reagents**. Human mesenchymal stem cells (*h*MSCs) were ordered from Lonza and used at passage 4 following manufacturer protocols. Standard MC3T3 fibroblasts were obtained from Riken and used at passage 6 following manufacturer protocols. The following reagents were used as received for cell culture and assessment of cellular activity: α-MEM, penicillin (10,000 U/mL)/streptomycin (10,000 μg/mL) (pen/strep), fetal bovine serum (FBS), trypsin blue, and the Live/Dead® assay kit were ordered from Life Technologies; trypsin-ethylene diamine tetraacetice acid (EDTA), Dulbecco's phosphate-buffered saline (PBS), 1,4-piperazinediethanesulfonic acid (PIPES), polyethylene glycol (PEG, 8000 kDa), paraformaldehyde, Triton™ X-100, sodium borohydride, donkey serum, and secondary donkey anti-mouse IgG-488 antibody were ordered from Fisher Scientific; ethylene glycol-*bis*(2-aminoethylether)-*N,N,N′,N′*-tetraacetic acid (EGTA), and mouse monoclonal primary anti-vinculin antibody were purchased from Sigma Aldrich; Rhodamine phalloidin was ordered from VWR; 4′,6-diamidino-2-pheny-lindole (DAPI) nuclear stain and the PrestoBlue® metabolic assay were ordered from Life Technologies Invitrogen™.

Ketamine HCl (KetaVed®, 100 mg/mL), Xylazine (AnaSed®, 20 mg/mL), Acepromazine Maleate (PromAce®, 10 mg/mL), Buprenorphine (Buprenex, 0.3 mg/mL), sodium pentobarbital (Beuthanasia®-D); Povidone-iodine solution (Vetadine); Modified Masson's Trichrome staining kit was ordered from Scytek Laboratories, Inc.; goat polyclonal primary anti-CD206 (C-20) antibody was ordered from SantaCruz Biotechnology; Mayer's Hematoxylin, Eosin Y, mouse monoclonal primary anti-CD68/SR-D1 (KP1) antibody, rabbit monoclonal primary anti-CCR7 (Y59) antibody, Sodium Citrate Dihydrate, DPX Mountant, Trizma® base (*Tris*(hydroxymethyl)aminomethane) (Acros Organics, 99.85%), Sodium Chloride (NaCl, Acros Organics), and Tween® 20 (Polyethylene glycol sorbitan monolaurate, Acros Organics) were ordered from Fisher Scientific; VECTASHIELD HardSet Mounting Medium was ordered from Vector Laboratories; donkey anti-mouse Alexa Fluor® IgG-350 secondary antibody (polyclonal, 2 mg/mL), donkey anti-goat Alexa Fluor® IgG-488 secondary antibody (polyclonal, 2 mg/mL), donkey anti-rabbit Alexa Fluor® IgG-546 secondary antibody (polyclonal, 2 mg/mL) and TRIS-HCl were ordered from Life Technologies Invitrogen™.

**In vitro characterization of cellular responses to degradable elastomers**
*Sample preparation and cell culture*. Samples for cell culture studies ($n = 5$) were prepared by spin-coating a solution of 0.4 *wt*% polymer in CHCl₃ on a glass coverslip (1 min at 1000 rpm). Films were spin-coated onto silicon wafers and glass coverslips to determine thickness by ellipsometry on a variable angle spectroscopic ellipsometer (VASE, M-2000 UV − visible−NIR [240−1700 nm] J. A. Woollam Co., Inc.). Angles used were 55–70 degrees in 5-degree increments, and the Cauchy Layer model was used to determine sample thickness, with all samples measuring *ca*. 60 nm. Spin-coated glass coverslips were then placed into 12 well plates for ethylene oxide (EtO) sterilization, using 0.5 cc/L EtO at room temperature and 35% humidity for 12 h with an Anprolene benchtop sterilizer (Anderson Products, Inc., Haw River, NC), followed by a 48 h purge.

Human Mesenchymal Stem Cells (*h*MSCs) and MC3T3 fibroblasts were expanded according to the manufacturer's protocol and cultured in α-MEM

supplemented with 10% FBS and 1% pen/strep in incubators maintained at 37 °C with 5% $CO_2$. Media was changed every day for the duration of culture.

*Cell viability*. Cell viability was assessed using a Live/Dead® assay kit. *h*MSCs (passage 4) and MC3T3 cells (passage 7) were seeded on spin coated coverslips ($n = 5$) at 4000 cells/cm². After 24 h the medium was removed and cells were stained using a Live/Dead® assay kit with 4 μM calcein AM (acetoxymethyl ester) and 2 μM ethidium homodimer-1 in PBS, and incubated in the dark at room temperature for 15 min before imaging using an Olympus IX81 inverted fluorescent microscope equipped with a Hamamatsu Orca R² fluorescent camera and Olympus CellSens® Dimension imaging software under TRITC (wavelength = 556/563 nm excitation/emission) and FITC (wavelength = 490/525 nm excitation/emission) channels to obtain 10 images at 10× magnification of each specimen. Live and dead cells were counted using NIH ImageJ. The number of live cells was divided by the total number of cells on each specimen to obtain a percentage of cell viability.

*Cell proliferation*. Cell proliferation of *h*MSCs seeded on spin-coated glass slides ($n = 4$, 1000 cells/cm²) was evaluated by metabolic activity using a PrestoBlue® Assay following the supplier's protocol. Metabolic activity was analyzed at 24 h, 3 days, and 7 days of culture. A standard curve was prepared by seeding cell suspensions at known concentrations into 12 well plates at least 6 h before the experiment to allow full attachment. Ten descending concentrations of cells obtained by serial dilution and one blank were included in the standard curve. After removing the medium, 1.5 mL of PrestoBlue® solution (10% in cell culture medium) was added to each well, followed by incubation at 37 °C for 2–4 h. Sample fluorescence was read when the fluorescence from the standard curve gave a linear fit. 100 μL of solution was taken from each well and placed in triplicate into a 96-well plate. The fluorescence intensity (FI) was detected in a BioTek® Synergy™ MX Microplate Reader at wavelengths of 570 nm for excitation and 615 nm for emission. The standard curve was fit with a linear relationship by plotting FI *vs* cell number, with a coefficient of determination ($R^2$) above 0.99. The cell number was approximated using the obtained equation.

*Cell seeding onto GRGDS functionalized polymer thin films*. Mouse calvarial stem cells (MC3T3-E1, Passage 10) were cultured using MEM α (Gibco, Life Technologies) supplied with 10 vol % FBS, 100 units/mL penicillin, and 100 μg/mL streptomycin at 37 °C and 5% $CO_2$. Cells were subcultured every 3 days with a 0.25% (w/v) trypsin and 0.5% (w/v) EDTA solution. Polymer films were sterilized by UV irradiation for 15 min, washed 3 times with PBS, and soaked with MEM α for 2 h prior to cell seeding. Cells were seeded onto polymer thin films at 25,000 cells/cm² ($n = 4$).

**In vivo characterization of tissue responses to degradable elastomers**. Elastomer samples were prepared using vacuum film compression (Technical Machine Products Corporation) to press 0.5 mm thick films at 110 °C under 2,000 lbs of pressure for one hour. PLLA samples were pressed at 200 °C for 10 min under 10,000 lbs of pressure, ten minutes under 15,000 lbs of pressure, and 10 min under 20,000 lbs of pressure. Elastomer samples all swelled after compression to be approximately 1 mm thick. PLLA films maintained 0.5 mm thickness. After cooling, films were cut into 8 mm diameter discs and placed into 12 well plates for ethylene oxide (EtO) sterilization. Two weeks prior to surgery, samples were sterilized with a dose of approximately 0.5 cc/L EtO at room temperature and 35% humidity for 12 h using an Anprolene benchtop sterilizer (Anderson Products, Inc.) followed by a 48 h purge.

Animals were handled and cared for in accordance with protocols that were approved by the University of Akron Institutional Animal Care and Use Committee. Female Sprague-Dawley rats (Harlan Laboratories) aged 60-80 days and weighing approximately 200-224 g were given one week to acclimate to the facility before performing surgeries. General anesthesia was induced using a cocktail of ketamine (29.6 mg/kg), xylazine (5.95 mg/kg) and acepromazine (0.53 mg/kg). Prior to surgery rats were also given 0.02 mg/kg Buprenorphine, which was administered again every 12 h as needed. The back was shaved and disinfected using several washes with povidone-iodine solution and sterile alcohol wipes. Four 1 cm incisions were made, each 1 cm away from the spine with at least 2 cm separating each incision to avoid sample crossover. A subcutaneous pocket was created using curved hemostats to tunnel into the fascia space anterior to the incision. Each polymer implant was placed into a pocket, and the incision was closed using Michel clips. An $n = 6$ for each polymer sample type was implanted per time point, with an $n = 12$ for the PLLA control in order to have one control sample per rat as direct comparison. The samples were randomized so that analyses could be performed to check for interactions between samples implanted within the same animal. Samples were implanted for 1 month, 2 months and 4 months.

Animals were euthanized using a fatal dose of sodium pentobarbital (0.5 cc per rat) at their respective time points. A midline incision was made along the spine of the rat and between each sample. Each sample was isolated, exposed, and photographed to observe macroscopic inflammation before being removed and fixed in 4% paraformaldehyde overnight. After fixation samples were rinsed for 15 min in distilled water three times, followed by three 15-min rinses in 70% ethanol. Samples were then processed into wax overnight using a tissue processor (Leica ASP300 S, Leica Biosystems) before embedding in paraffin wax for sectioning.

*Histology staining and imaging.* Sections (8 to 14 μm thick) were stained for brightfield imaging with hematoxylin and eosin (H&E), and modified Masson's trichrome. All images were taken with a VS120-S6-W automated microscope equipped with both a CCD color camera and a fluorescence Hamamatsu Orca-Flash4.0 fluorescence camera using DAPI (ex/em = 350/470 nm), FITC (ex/em = 490/525 nm), and TRITC (ex/em = 556/563 nm) filters. Brightfield images were analyzed for inflammatory markers including granuloma thickness, and qualitatively assessed for general inflammation compared to PLLA control samples. H&E slides were also analyzed under 400× light microscopy for a number of inflammatory cells by a board-certified veterinary pathologist utilizing a modified scoring system designed by the International Organization for Standardization (ISO 10993-6 Annex E). Scoring was based on a scale from 0 to 4 (0 = none; 1 = Rare, 1-5 Minimal; 2 = 5–10, Mild; 3 = Heavy Infiltrate, Moderate; 4 = Packed, Severe). The Macrophage analysis from immunofluorescent images included qualitative assessment of cells located within the inflammatory region surrounding the implants where CD68 was a positive indicator of a macrophage, CCR7 indicated primarily M1 expression, and CD206 indicated M2 expression. Samples that showed tissue infiltration into the polymer space were stained with picrosirius red to observe collagen deposition and orientation. Images were taken using an Olympus IX70 microscope equipped with a camera at 40× magnification under brightfield and polarized light conditions using Olympus MicroSuite™ imaging software.

**Swelling tests**. A film (0.5 mm thickness) of each elastomer was prepared from vacuum film compression using the same method as stated above. Discs (4 mm) were cut from the film and placed in 1× PBS in the incubator (37 °C, 5% $CO_2$ humidified atmosphere) for up to 32 days. The swelling behavior of the elastomers were determined by tracking the wet and dry mass of the disc samples ($n = 3$) at each time point.

**Statistics**. Results are reported as mean ± standard deviation. One-way analysis of variance (ANOVA) with Tukey's post-hoc was performed with a 95% confidence.

## Data availability
All raw spectroscopic and histology data will be made available upon request.

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

## Acknowledgements
We are grateful for financial support from the Biomaterials Division of the National Science Foundation (DMR-1507420), the W. Gerald Austen Endowed Chair in Polymer

Science and Polymer Engineering via the John S. and James L. Knight Foundation (MLB), ERC Grant (Number 681559) (APD), and the National Health and Medical Research Council (NHMRC) of Australia (APP 1054569) (CAB). The authors would like to thank Gina M. Policastro for monomer synthesis, James A. Wilson for polymer precursor synthesis, Derek Luong for help with in vitro assessments, Dr. Christopher Klonk and Dr. Christopher Premanandan for assistance with histological assessments.

## Author contributions

A.P.D. and M.L.B. conceived the project idea. C.A.B., A.P.D., and M.L.B. designed the materials and synthetic routes while C.A.B., A.P.B., and Y-H.H. synthesized and characterized the materials. C.B. and J.Y. performed thermal, mechanical analyses and in vitro degradation studies. M.B.W. and M.L.B designed the in vitro and in vivo experiments. M.C.A., N.Z.D., Y-H. H. and M.B.W. prepared samples and performed in vitro analyses. M.B.W., M.L.B., and N.Z.D. prepared samples, performed in vivo analyses, and performed the histology. A.P.D. and M.L.B. wrote the manuscript, all authors edited and commented on the manuscript.

## Competing interests

A patent application was submitted in 2018 by M.L.B. and A.P.D. covering some aspects of this work.
