## [Peer Review File · Nature Communications]

Reviewers' comments:

Reviewer #1 (Remarks to the Author):

The present paper shows the synthesis of novel elastomers and their polymerisation in to biodegradable biomaterials with mechanical properties tuned to their degradation. The paper also assesses the in vitro and in vivo biocompatibility of the synthesised biomaterials. The work of this paper emerges for the innovative character of the biomaterial synthesis that is supported by a sound hypothesis, i.e. the need to control the mechanical properties of the material upon degradation. This is particularly commendable if considered that most of the research in this field is based on the engineering of well known biomaterials that have not been able to fulfil this requirement.

The hypothesis of the work is that these properties can be beneficial in regenerative medicine. This is a rather vague statement as regenerative medicine is defined in three main therapeutic approaches (tissue engineering, cell therapy and gene therapy) and the biomaterial properties for any of these types of treatments have to be tailor made for specific clinical applications. The work of this paper seems to be suitable for tissue engineering, but the potential clinical applications have not been identified. Hence, the assessment of both the physicochemical properties and biocompatibility properties of the biomaterials is difficult to make.

The experimental work has been performed in a very robust and reliable manner, particularly that of material synthesis. However, the choice of some of the characterisation methods has not always been appropriate to test the work hypothesis. In particular, the mechanical properties have been performed on vacuum-compressed samples by Instron in a dry state that does not reflect the wet conditions of the in vivo environment. The swelling behaviour tests, performed on specimens prepared by the same method, show different profiles of mass change when dry, wet and swollen materials are tested. Hence, it can be argued that the testing of the mechanical properties of swollen specimens would have been more appropriate to reflect the response to the in vivo microenvironment. While it is recognised that the testing of the dry specimens is still valuable as it shows the differences that subtle changes can make on the mechanical properties, the results obtained do not offer any indication of the material suitability for specific clinical applications. As later demonstrated by the in vivo experiments, the increased percentage of succinate in the polymer leads to the infiltration of tissue. However, in the in vitro accelerated degradation study the SEM images and their relative description of the results focus on the surface erosion without showing any data relative to the bulk erosion. Additional SEM cross-section images of the samples immersed in the alkaline medium would have provided valuable information to link the physicochemical characterisation to the in vivo resorption patterns.

As far as the in vitro and in vivo biocompatibility tests are concerned, these are significant weakness of the work. As stated above, the experiments have been performed in a robust, competent and reliable manner, but the choice of the experimental design and the results are not addressing the work hypothesis satisfactorily. The choice of using mesenchymal stem cells for the in vitro biocompatibility tests is not justified unless the authors indicate that the aim of the work is to produce biomaterial as support for stem cells in tissue engineering applications. In vitro biocompatibility tests should be conducted using cells relevant to the clinical application(s) for which the biomaterial has been designed for and in relation to the host response to the implant. In this respect, and retrospectively looking at the in vivo data, it would have been more appropriate to test the material in vitro using fibroblasts and inflammatory cells (i.e.

monocytes/macrophages). In any case, the work performed on stem cells is very superficial and show that the biomaterial produced are not an ideal substrate for this type of cells unless the GRGDS bioligand is grafted to it. The stem cells have been characterised for their morphology, but not for their markers of multi-potency. Hence, it is not clear if they are able to preserve their regenerative potential or rather differentiate in to fibroblast-like cells as the morphology suggests. The assessment of the in vivo data clearly shows the formation of a fibrotic capsule that, in a clinical application, would lead to the isolation of the biomaterials rather than its integration into the tissue. The observation that the biomaterial biocompatibility can be considered acceptable if the thickness of the capsule does not exceed a certain value is a very old, surpassed concept for biomedical implants and not suitable at all for regenerative medicine applications. In the case of

the use of this class of polymers for the manufacturing of medical implants, the formation of the fibrotic capsule would lead to the isolation of the implant rather than its integration into the treated tissue. This view is also informed by the analysis of the specimens retrieved at different times of implantation (see supplementary material). Here, the visual inspection clearly shows that the specimens are very clean and only some angiogenesis can be observed on the surface of the implanted discs. This means that there was no significant integration with the surrounding fibrotic capsule and that in a clinical application the implant would be loose and mechanically unstable. An integrated biomaterial would have been difficult to separate from the capsule upon explantation. Such a loose integration is the main cause of aseptic failure of implants. In the case of tissue engineering applications (i.e. regenerative medicine) the formation of the capsule would be a barrier to tissue regeneration. In both cases, the stabilisation of the capsule that appears in all the formulations tested after 4 months is likely to lead to the formation of a tissue gap rather than tissue regeneration when the biomaterial will be fully degraded. This is a problem well known in literature with biomaterials that degrades slowly and that are accompanied by the formation of a fibrotic capsule. Hence, this paper shows no real progress. From the point of view of the study design, it is not clear why the authors decided to implant biomaterials that had not been functionalised with the GRGDS as these were the ones showing better cell substrate properties in vitro that could have eliminated the formation of the fibrotic capsule and improved the integration in to the surrounding tissue. As far as the host response to the implant is concerned, it is clear that this is different in the various formulations tested. An in vitro study with inflammatory cells as that suggested above, would have provided valuable indications about the formulations and driven the selection of the types of biomaterials to be tested in vivo.

In summary, while this paper is commendable in its attempt to show the potential of tuning resorption and mechanical properties of polymeric biomaterials, it does not offer sufficient indications about its potential clinical application and/or a step change in the resolution of the limitations affecting the biocompatibility of most biomaterials.

Reviewer #2 (Remarks to the Author):

This manuscript describes a series of polymers produced by thiol-yne addition. The authors show an effective control of modulus by altering cis:trans ratio of the double bonds in the backbone of the polymer. When using a monomer with an extra alkyne group, the polymer can be further modified with azide-containing peptides by click chemistry. The polymer appears to be well tolerated in a SC implantation model in rodents. The study is in general well planned and the polymer synthesis route is flexible and well designed. The study could be strengthened significantly by considering the following factors-

1. The cis-trans ratio is the key to the polymers' property, yet the manuscript had no information on how the ratio is controlled. The author glossed over the detailed with a sweep statement of "judicious choice of polymerization catalyst and solvent...". It is therefore impossible to determine how this is achieved and what mechanism might be at play.
2. The molecular weight of some of the polymers used to compare modulus is too different to be appropriate. Molecular weight has a fundamental impact on a polymer's mechanical property.
3. It would help if the authors could provide certain level of explanation on why the cis:trans ratio has a big effect on mechanical properties and degradation rate. As it is now, the manuscript stays at a descriptive level with low mechanistic insight.
4. To definitively show that polymer is an elastomer, hysteresis data need to be presented with cycles at least over 20 to demonstrate low hysteresis.
5. How was modulus defined and measured? What range of the stress-strain curve was used? Since no detailed data was provided, a simple estimate based on the UTS and strain doesn't support the claimed 15-fold change.
6. The author should conduct a proper cytotoxicity study with proper controls. This could be a standard live-dead assay, or a cell metabolic activity assay with TCPS as a control or another surface commonly used in cell culture.
7. Early time points should be included to assess acute inflammation. The implants seemed to have been removed from the explant? Or was it tissue processing artifact? The polymer implants should be kept in place to ensure consistency across different samples and a complete

representation of the polymer-tissue interface. Higher magnification micrographs should be provided to assess the inflammatory response.

8. Statement on P. 12: "...degrading in vivo over a period of four months and were replaced with mature and developing tissues." Is confusing, please clarify what the mature and developing tissue was referring to.

Reviewer #3 (Remarks to the Author):

This study is significant as it shows a novel way to control mechanical and degradation properties independently of each other. Most of the previous approaches have been based on composition control and blending of different materials, which result in concurrent shifts of many physical properties. In this paper, the authors used a very elegant synthetic approach by altering the stoichiometry of succinate incorporation to precisely regulate the degradation rate of the material while retaining control over the mechanical properties by maintaining the cis/trans stereochemistry of the double bond. The iso-compositional control of different physical properties (e.g., mechanics and degradation) is vital for the design of biomedical devices and tissue engineering. In this respect, the distinctly low inflammatory response of the synthesized materials is very encouraging.

The paper is recommended for publication in Nature Materials after addressing a few technical issues:

1) The beginning of sentence 1 in the Abstract is misleading ("biological tissues are highly elastic in nature"), because elasticity is not the only property that should be replicated. It also contradicts the first sentence in the Introduction: "Biological tissues are highly viscoelastic and dynamic in nature."

The authors may want to simply start with "Efforts to repair or replace cartilage, tendon, muscle, and vasculature have been ongoing for decades." Also, abstract should be slightly modified to highlight the novelty and potential impact of this study.

2) Some of the mechanical curves display distortions at higher stress values. This might be due to sample slipping out of clamps and is easy to fix.

3) The low-strain section of the curves suggests yielding behavior. Are these materials elastic? Is deformation reversible?

Author's Response to Reviewer #1:

The present paper shows the synthesis of novel elastomers and their polymerisation in to biodegradable biomaterials with mechanical properties tuned to their degradation. The paper also assesses the in vitro and in vivo biocompatibility of the synthesised biomaterials. The work of this paper emerges for the innovative character of the biomaterial synthesis that is supported by a sound hypothesis, i.e. the need to control the mechanical properties of the material upon degradation. This is particularly commendable if considered that most of the research in this field is based on the engineering of well known biomaterials that have not been able to fulfil this requirement.

Thanks for recognizing the key point. There is a significant gap in the availability of resorbable, elastic materials for tissue engineering. Very few new systems are introduced with translational relevance and we feel strongly that these materials will have lasting impact especially given the ability to tune resorption rates.

The hypothesis of the work is that these properties can be beneficial in regenerative medicine. This is a rather vague statement as regenerative medicine is defined in three main therapeutic approaches (tissue engineering, cell therapy and gene therapy) and the biomaterial properties for any of these types of treatments have to be tailor made for specific clinical applications. The work of this paper seems to be suitable for tissue engineering, but the potential clinical applications have not been identified. Hence, the assessment of both the physicochemical properties and biocompatibility properties of the biomaterials is difficult to make.

We agree with the reviewer and their comment of a truly vague statement. Like we noted above, there is a significant gap in the availability of resorbable, elastic materials for tissue engineering broadly. We don't ultimately know which application these materials might be ultimately used in. This work was included as first report of what we believe to be a new class of surface-eroding elastomers. There are several ongoing *in vivo* works on cartilage, volumetric muscle and tendon regeneration with these materials. They will be reported as appropriate in the future. However, in this communication we aim to report the breadth of mechanical properties while being able to design for degradation. The ability to control both materials in stoichiometrically identical materials is unprecedented and ultimately what we feel is valuable to the reader in this initial report.

The experimental work has been performed in a very robust and reliable manner, particularly that of material synthesis. However, the choice of some of the characterisation methods has not always been appropriate to test the work hypothesis. In particular, the mechanical properties have been performed on vacuum-compressed samples by Instron in a dry state that does not reflect the wet conditions of the in vivo environment. The swelling behaviour tests, performed on specimens prepared by the same method, show different profiles of mass change when dry, wet and swollen materials are tested. Hence, it can be argued that the testing of the mechanical properties of swollen specimens would have been more appropriate to reflect the response to the in vivo microenvironment.

We show the swelling results (Figure S26) to demonstrate that there is virtually no swelling (<1%) of the materials. This is also true at 37 C. The elastomeric materials are already above the glass transition. Therefore, the mechanical properties are not any different at 37 C than they are at the ambient temperature conditions. This stands in stark contrast to the vast majority of degradable polyesters where the materials swell and exhibit bulk erosion characteristics which leads to rapid loss of properties and if sufficient volume exists, acidosis of surrounding tissues. Given the current state of our laboratories which will likely be down for 3-6 addition months following my move to Duke University, temperature dependent mechanical properties are simply not possible at this time.

While it is recognised that the testing of the dry specimens is still valuable as it shows the differences that subtle changes can make on the mechanical properties, the results obtained do not offer any indication of the material suitability for specific clinical applications.

We have included some comments about what material might be used for clinically. However, clinical applications will be defined in the future after additional investigations. This contribution has been intended as a materials-focused paper that is not specifically target toward an application. We anticipate the unique elastomeric and degradation properties will be of significant interest to the tissue engineering community.

As later demonstrated by the *in vivo* experiments, the increased percentage of succinate in the polymer leads to the infiltration of tissue. However, in the *in vitro* accelerated degradation study the SEM images and their relative description of the results focus on the surface erosion without showing any data relative to the bulk erosion.

As we show in our data, there is virtually no swelling and therefore, no bulk erosion processes are ongoing. This is one of the most significant findings in this class of material and the surface erosion properties affords the ability to design for degradation. This is distinctly different that degradable polyesters and polyurethanes.

Additional SEM cross-section images of the samples immersed in the alkaline medium would have provided valuable information to link the physicochemical characterisation to the *in vivo* resorption patterns.

See note above. The SEM cross sections look identical to the bulk amorphous material. Since the materials are not swelling, there is no change in the appearance of the bulk properties.

As far as the *in vitro* and *in vivo* biocompatibility tests are concerned, these are significant weakness of the work. As stated above, the experiments have been performed in a robust, competent and reliable manner, but the choice of the experimental design and the results are not addressing the work hypothesis satisfactorily. The choice of using mesenchymal stem cells for the *in vitro* biocompatibility tests is not justified unless the authors indicate that the aim of the work is to produce biomaterial as support for stem cells in tissue engineering applications. *In vitro* biocompatibility tests should be conducted using cells relevant to the clinical application(s) for which the biomaterial has been designed for and in relation to the host response to the implant. In this respect, and retrospectively looking at the *in vivo* data, it would have been more appropriate to test the material *in vitro* using fibroblasts and inflammatory cells (i.e. monocytes/macrophages). In any case, the work performed on stem cells is very superficial and show that the biomaterial produced are not an ideal substrate for this type of cells unless the GRGDS bioligand is grafted to it.

The viability of standard NIH 3T3 fibroblasts has been included and is near quantitative (added to Figure 4). In the revised manuscript we have included data for these well established fibroblasts. The tissue engineering community is interested in the stem cell populations so we felt inclusion would provide some level of comfort that they could be utilized in these and future experiments.. As their integrin mediated interactions are not as distinct as fibroblasts, it is not surprising that they would not adhere without significant serum adsorption or incorporation of integrin specific ligands.

The stem cells have been characterised for their morphology, but not for their markers of multi-potency. Hence, it is not clear if they are able to preserve their regenerative potential or rather differentiate in to fibroblast-like cells as the morphology suggests.

At early time points, they retain phenotypic morphology but over time, they differentiate into fibroblastic lineages. The point of the included experiments to show the materials are non-toxic. This is further reflected in the additional MC3T3 fibroblast viability assays. In the future, we plan to target specific applications with more refined and application specific *in vitro* assays.

The assessment of the *in vivo* data clearly shows the formation of a fibrotic capsule that, in a clinical application, would lead to the isolation of the biomaterials rather than its integration into the tissue. The observation that the biomaterial biocompatibility can be considered acceptable if the thickness of the capsule does not exceed a certain value is a very old, surpassed concept for biomedical implants and not suitable at all for regenerative medicine applications.

The capsule thickness is indeed a more traditional but also trusted and widely accepted measurement in the histopathology community. To this day it is used routinely to benchmark new materials against traditional materials, especially for regulatory agencies such as the US Food and Drug Administration. When these materials are fabricated into porous designs for specific tissue engineering applications, we agree that there would be more appropriate models for gauging the host tissue response.

In the case of the use of this class of polymers for the manufacturing of medical implants, the formation of the fibrotic capsule would lead to the isolation of the implant rather than its integration into the treated tissue. This view is also informed by the analysis of the specimens retrieved at different times of implantation (see supplementary material). Here, the visual inspection clearly shows that the specimens are very clean and only some angiogenesis can be observed on the surface of the implanted discs. This means that there was no significant integration with the surrounding fibrotic capsule and that in a clinical application the implant would be loose and mechanically unstable. An integrated biomaterial would have been difficult to separate from the capsule upon explantation. Such a loose integration is the main cause of aseptic failure of implants.

The faster degrading materials do not form a significant capsule as we note. This has been clarified in the text. When these materials are fabricated into porous designs for specific tissue engineering applications, there will be tissue integration, especially when functionalized with adhesive or homing peptide sequences. If there is no swelling of the polymer, there is no path for hydrolytic degradation to occur and limited opportunities for cellular and tissue penetration.

In the case of tissue engineering applications (i.e. regenerative medicine) the formation of the capsule would be a barrier to tissue regeneration. In both cases, the stabilisation of the capsule that appears in all the formulation tested after 4 months is likely to lead to the formation of a tissue gap rather than tissue regeneration when the biomaterial will be fully degraded. This is a problem well known in literature with biomaterials that degrades slowly and that are accompanied by the formation of a fibrotic capsule. Hence, this paper shows no real progress.

Most synthetic materials will form an initial capsule but this is often remodeled over time depending on the material. In this particular case, the materials are inducing the expected response. With a less significant inflammatory response, the surrounding tissue should continue to regenerate. Longer resorption times are seen in the formation of a capsule while the materials which erode faster are not.

From the point of view of the study design, it is not clear why the authors decided to implant biomaterials that had not been functionalised with the GRGDS as these were the ones showing better cell substrate properties in vitro that could have eliminated the formation of the fibrotic capsule and improved the integration in to the surrounding tissue.

A more extended tissue specific experiment is ongoing. However, as noted above, this contribution is trying to highlight the utility and potential of the materials. We do not wish to dilute the initial impact of a new family resorbable elastomers. They do show the enhanced integration noted as one would expect in early timepoints.

As far as the host response to the implant is concerned, it is clear that this is different in the various formulations tested. An in vitro study with inflammatory cells as that suggested above, would have provided valuable indications about the formulations and driven the selection of the types of biomaterials to be tested in vivo.

A more extended tissue specific experiment is ongoing. We agree with the reviewers assessment and will be measuring inflammation specifically in future studies as the response would strongly depend on the site of implantation.

In summary, while this paper is commendable in its attempt to show the potential of tuning resorption and

mechanical properties of polymeric biomaterials, it does not offer sufficient indications about its potential clinical application and/or a step change in the resolution of the limitations affecting the biocompatibility of most biomaterials.

Respectfully, there are no surface eroding elastomeric materials in the literature that do not rely on a hard block-soft block interaction which dictate its mechanical and resorption properties.

Author's Response to Reviewer #2 :

This manuscript describes a series of polymers produced by thiol-yne addition. The authors show an effective control of modulus by altering cis:trans ratio of the double bonds in the backbone of the polymer. When using a monomer with an extra alkyne group, the polymer can be further modified with azide-containing peptides by click chemistry. The polymer appears to be well tolerated in a SC implantation model in rodents. The study is in general well planned and the polymer synthesis route is flexible and well designed. The study could be strengthened significantly by considering the following factors-

1. The cis-trans ratio is the key to the polymers' property, yet the manuscript had no information on how the ratio is controlled. The author glossed over the detailed with a sweep statement of "judicious choice of polymerization catalyst and solvent...". It is therefore impossible to determine how this is achieved and what mechanism might be at play.

We have included a mechanistic explanation of the effect and have pulled the data table and the NMR back into the main text from the SI. Hopefully, this will clarify the role of the solvent and the catalyst.

2. The molecular weight of some of the polymers used to compare modulus is too different to be appropriate. Molecular weight has a fundamental impact on a polymer's mechanical property.

The molecular weight of entanglement for these materials is near 15 kDa which is where the material's thermal and mechanical properties stabilize. Therefore, at higher molar mass, the mechanical properties are not significantly different.

3. It would help if the authors could provide certain level of explanation on why the cis:trans ratio has a big effect on mechanical properties and degradation rate. As it is now, the manuscript stays at a descriptive level with low mechanistic insight.

The variations in mechanical properties correlate with the % cis stereochemistry in the same mechanism as natural rubber. There is a self association of the Cis domains in the materials that can be considered as physical crosslinking. The text has been modified to reflect this.

4. To definitively show that polymer is an elastomer, hysteresis data need to be presented with cycles at least over 20 to demonstrate low hysteresis.

We are unable to present extended hysteresis data at this time due to my move to Duke followed by what is expected to be an extended shutdown due to the global Covid19 pandemic.

5. How was modulus defined and measured? What range of the stress-strain curve was used? Since no detailed data was provided, a simple estimate based on the UTS and strain doesn't support the claimed 15-fold change.

These data have been included in the revision.

6. The author should conduct a proper cytotoxicity study with proper controls. This could be a standard live-dead assay, or a cell metabolic activity assay with TCPS as a control or another surface commonly used in cell culture.

These have been included using standard fibroblast cells as requested. The viability is nearly quantitative.

7. Early time points should be included to assess acute inflammation. The implants seemed to have been removed from the explant? Or was it tissue processing artifact? The polymer implants should be kept in place to ensure consistency across different samples and a complete representation of the polymer-tissue interface. Higher magnification micrographs should be provided to assess the inflammatory response.

The polymers were lost in the xylene dehydration step of histology processing. This was not expected.

8. Statement on P. 12: "...degrading in vivo over a period of four months and were replaced with mature and developing tissues." Is confusing, please clarify what the mature and developing tissue was referring to.

We have removed mature from the comments.

Author's Response to Reviewer #3:

This study is significant as it shows a novel way to control mechanical and degradation properties independently of each other. Most of the previous approaches have been based on composition control and blending of different materials, which result in concurrent shifts of many physical properties. In this paper, the authors used a very elegant synthetic approach by altering the stoichiometry of succinate incorporation to precisely regulate the degradation rate of the material while retaining control over the mechanical properties by maintaining the cis/trans stereochemistry of the double bond. The iso-compositional control of different physical properties (e.g., mechanics and degradation) is vital for the design of biomedical devices and tissue engineering. In this respect, the distinctly low inflammatory response of the synthesized materials is very encouraging.

The paper is recommended for publication in Nature Materials after addressing a few technical issues:

1) The beginning of sentence 1 in the Abstract is misleading ("biological tissues are highly elastic in nature"), because elasticity is not the only property that should be replicated. It also contradicts the first sentence in the Introduction: "Biological tissues are highly viscoelastic and dynamic in nature."

The authors may want to simply start with "Efforts to repair or replace cartilage, tendon, muscle, and vasculature have been ongoing for decades." Also, abstract should be slightly modified to highlight the novelty and potential impact of this study.

We have altered the abstract and text accordingly.

2) Some of the mechanical curves display distortions at higher stress values. This might be due to sample slipping out of clamps and is easy to fix.

This is due to necking of the tensile samples. There was no slippage.

3) The low-strain section of the curves suggests yielding behavior. Are these materials elastic? Is deformation reversible?

There is some yielding in the low strain regions and some hysteresis. We still field elastomer is an appropriate description. A more complete description will be included in future studies.

4) What is the crosslink density of these elastomers? Does it correlate with the modulus?

These materials are not chemically crosslinked which makes them significantly different than other materials in the literature. The variations in mechanical properties correlate with the % cis stereochemistry in the same mechanism as natural rubber. There is a self association of the Cis domains in the materials that can be considered as physical crosslinking.

REVIEWER COMMENTS

Reviewer #2 (Remarks to the Author):

The authors addressed several concerns but brushed aside others, including two major concerns. The stress-strain curves show soft materials that neck at high stress regions. No data is presented that support the elastomer claim, which is central to this paper. The motivation of the materials is biological tissue and the intended application is medical. However, compared with PLLA control, the host mounts a highly aggressive inflammatory response, as shown in Fig 5 G, I, J, and K. The authors state that the polymer is fully degraded in Fig 5K, yet there is still severe inflammation. This will likely lead to scarring. Related to this, fibrous encapsulation can place significant limitations on the utility of the materials. The biocompatibility test didn't include collagen deposition at all.

Reviewer #3 (Remarks to the Author):

My previous comments were not addressed. Therefore, I repeat them:

- 1) The opening sentence in the Abstract ("biological tissues are highly elastic in nature") is misleading and contradicts to the first sentence of the Introduction: "Biological tissues are highly viscoelastic and dynamic in nature."
- 2) Unlike tissues, the reported materials are inelastic since they do not restore their original shape upon unloading.
- 3) These materials cannot be called elastomers since they are not crosslinked and demonstrate the characteristic yielding behavior of conventional plastics.
- 4) Some of the mechanical curves have an erratic shape, which questions the reliability of the UTS measurements. High frequency oscillations and sudden stress upsurges are not due to necking.
- 5) Initial slopes of the 9 and 18% curves suggest very high Young's modulus, much higher than ~30MPa indicated in table 2. How was the modulus measured?

Author's Response to Reviewer #2:

The authors addressed several concerns but brushed aside others, including two major concerns. The stress-strain curves show soft materials that neck at high stress regions. No data is presented that support the elastomer claim, which is central to this paper.

We agree that some of the materials neck at extremely high strain. This is not surprising given that this phenomena generally occur at > 1500 percent strain. Unlike conventional crosslinked or vulcanized elastomers which possess covalent bonds which prove toughness, the elastic components of these materials arise from the aggregation of the cis alkene bonds and function in a similar manner to an ABA triblock copolymer where isolated domains contribute to elasticity. We have included cyclic tensile testing data in the SI (and below) to show elastic behavior. While some fatigue is evident after multiple cycles, we believe the behavior is consistent with elastomeric properties.

Figure S26. The hysteresis was performed by load-unloaded cyclic stress vs. strain curves stretching up to 100% with 5 cycles at 10 mm/min strain rate.

The motivation of the materials is biological tissue and the intended application is medical. However, compared with PLLA control, the host mounts a highly aggressive inflammatory response, as shown in Fig 5 G, I, J, and K.

At 4 months the crystalline PLLA is really not degrading and therefore one would expect little inflammation other than from the small quantities of acidic byproducts. As shown in what will become part of the new Figure 5, M0 and M1 macrophages are visible in the PLLA samples at 4 months. See elaboration below.

The authors state that the polymer is fully degraded in Fig 5K, yet there is still severe inflammation. This will likely leads to scarring. Related to this, fibrous encapsulation can place significant limitations on the utility of the materials. The biocompatibility test didn't include collagen deposition at all.

There is no doubt inflammation during the degradation and resorption processes but in the macrophage phenotype staining shown above and below, M₀, M₁ and M₂ macrophages can each be found. According to the traditional concept, macrophages are classified into pro-inflammatory (M₁), non-activated (M₀) or anti-inflammatory (M₂) subsets that play distinct roles in the initiation and resolution of inflammation. The slight autofluorescence of the polymer prevents distinct quantification of the relative macrophage concentrations. However, coupled with the collagen deposition data, we feel this is compelling indications of resorption and remodeling.

While one hope to always do more, these latest experiments have now exhausted our histology specimens. We will be looking at this more quantitatively in the future with more translationally-focused pre-clinical applications.

The H&E and more importantly the trichrome images shown below in what is now a new **Figure 5**, show no indication of scarring or fibrous encapsulation. While there is collage deposition around all the implants at early timepoints, the stereoelastomers have significantly less than the PLLA controls at each timepoint.

Picrosirius red (PSR) staining is a commonly used histological technique to visualize collagen in paraffin-embedded tissue sections. PSR stained collagen appears red in optical microscopy. However, it is largely unknown that PSR stained collagen also shows a red fluorescence, whereas live cells have a distinct green autofluorescence. As shown below in what is now Figure 6, Picrosirius Red staining is present at the edge of the degrading polymer (1 month) and throughout the site formerly occupied by the degraded 80cis100suc elastomers (4 months) of subcutaneous incubation. This clearly shows that collagen deposition and maturation occurred throughout the space formerly occupied by the polymer.

Author's Response to Reviewer #3:

My previous comments were not addressed. Therefore, I repeat them:

1) The opening sentence in the Abstract (“biological tissues are highly elastic in nature”) is misleading and contradicts to the first sentence of the Introduction: “Biological tissues are highly viscoelastic and dynamic in nature.”

This has been corrected to be consistent. I am sorry for the previous oversight.

2) Unlike tissues, the reported materials are inelastic since they do not restore their original shape upon unloading.

Thanks for your concerns. We acknowledge that the materials are not fully elastic as they show some hysteresis. We took 49% succinate with 79% *cis* as an example to perform cyclic tests (5 cycles) up to 100% to show the hysteresis. The figure shown below has been added into supporting information section. Like we note below, while some fatigue is evident after multiple cycles, we believe the behavior is consistent with elastomeric properties. We call these materials “elastomers” based on their projected use in tissue engineering application. We showed the full spec of stress vs. strain curves; however, typically natural tissues would never be stretched more than 100% of its original shape.

3) These materials cannot be called elastomers since they are not crosslinked and demonstrate the characteristic yielding behavior of conventional plastics.

Unlike conventional crosslinked or vulcanized elastomers which possess covalent bonds which prove toughness, the elastic components of these materials arise from the aggregation of the *cis* alkene bonds and function in a similar manner to an ABA triblock copolymer where isolated domains contribute to elasticity. We have included cyclic tensile testing data in the SI to show elastic behavior. While some fatigue is evident after multiple cycles, we believe the behavior is consistent with elastomeric properties. We call these materials “elastomers” based on their projected use in tissue engineering application. We showed the full spec of stress vs. strain curves; however, typically natural tissues would never be stretched more than 100% of its original shape.

4) Some of the mechanical curves have an erratic shape, which questions the reliability of the UTS measurements. High frequency oscillations and sudden stress upsurges are not due to necking.

We considered the tensile testing instrumentation limitations (from manual grips not air pressure grips) resulted in samples being slippery at region of very high strain% to cause these erratic shapes at some positions. This has not altered the measurement in a significant way.

5) Initial slopes of the 9 and 18% curves suggest very high Young's modulus, much higher than ~30MPa indicated in table 2. How was the modulus measured?

The Young's modulus calculation based on the initial 1%-5% strain%, where is the linear region.

REVIEWERS' COMMENTS

Reviewer #2 (Remarks to the Author):

The additional data on inflammation is sufficient considering we are in the middle of a pandemic. The new data on cyclic test of the polymer is very informative. However, it demonstrates that the polymer is not an elastomer. It is true that 'elastomers' have a range of elastic recoils, some have more hysteresis than others. It is also true that there is no clear demarcation on how small the hysteresis needs to be for one to be called an elastomer. When the data (Fig S26) shows 60% plastic deformation at cycle 5 when the strain is at 100%, I would not call it an elastomer. In the grand scheme of things, the value of this research is the novel use of cis/trans conformation in controlling mechanical properties. Whether the product is an elastomer or not is a minor point. I would be happy to suggest publishing the work if the author is willing to call the polymer a soft stretchy polymer. In this case, calling "it is what it is" is correct. This is a data-rich manuscript. Much work went into it. I hate to keep beating the same drum, but I must respectfully disagree with calling this an elastomer.

Reviewer #3 (Remarks to the Author):

Naming these materials as elastomers is highly misleading. This term should be removed and replaced.

These materials undergo yielding at low strains. Their deformation is irreversible. Their modulus is very high. None of these behaviors pairs with elastomers. A high-density PE sample would show similar stress-strain curves with nearly identical modulus values. Crystallites in HDPE play role of crosslinkers. However, the deformation mechanism is completely different from elastomers and therefore nobody would call them as such.

Author's Response to Reviewer #2:

The additional data on inflammation is sufficient considering we are in the middle of a pandemic. The new data on cyclic test of the polymer is very informative. However, it demonstrates that the polymer is not an elastomer. It is true that 'elastomers' have a range of elastic recoils, some have more hysteresis than others. It is also true that there is no clear demarcation on how small the hysteresis needs to be for one to be called an elastomer.

When the data (Fig S26) shows 60% plastic deformation at cycle 5 when the strain is at 100%, I would not call it an elastomer. In the grand scheme of things, the value of this research is the novel use of cis/trans conformation in controlling mechanical properties. Whether the product is an elastomer or not is a minor point. I would be happy to suggest publishing the work if the author is willing to call the polymer a soft stretchy polymer. In this case, calling "it is what it is" is correct. This is a data-rich manuscript. Much work went into it. I hate to keep beating the same drum, but I must respectfully disagree with calling this an elastomer.

I agree and we have changed the nomenclature throughout the manuscript to elastomer-like as suggested by the editor, stereomers or plastic.

Author's Response to Reviewer #3:

Naming these materials as elastomers is highly misleading. This term should be removed and replaced.

These materials undergo yielding at low strains. Their deformation is irreversible. Their modulus is very high. None of these behaviors pairs with elastomers. A high-density PE sample would show similar stress-strain curves with nearly identical modulus values. Crystallites in HDPE play role of crosslinkers. However, the deformation mechanism is completely different from elastomers and therefore nobody would call them as such.

I agree and we have changed the nomenclature throughout the manuscript to elastomer-like as suggested by the editor, stereomers or plastic.